# Purine and pyrimidine synthesis differently affect the strength of the inoculum effect for aminoglycoside and β-lactam antibiotics

Daniella M. Hernandez,[1] Melissa Marzouk,[1,2] Madeline Cole,[3] Marla C. Fortoul,[3] Saipranavi Reddy Kethireddy,[2] Rehan Contractor,[2] Habibul Islam,[4] Trent Moulder,[2] Ariane R. Kalifa,[1,2] Estefania Marin Meneses,[1,2] Maximiliano Barbosa Mendoza,[1] Ruth Thomas,[2] Saad Masud,[3] Sheena Pubien,[3] Patricia Milanes,[3] Gabriela Diaz-Tang,[1,2] Allison J. Lopatkin,[4,5,6] Robert P. Smith[1,3]

**ABSTRACT**  The inoculum effect has been observed for nearly all antibiotics and bacterial species. However, explanations accounting for its occurrence and strength are lacking. Previous work found that the relationship between [ATP] and growth rate can account for the strength and occurrence of the inoculum effect for bactericidal antibiotics. However, the molecular pathway(s) underlying this relationship, and therefore determining the inoculum effect, remain undiscovered. Using a combination of flux balance analysis and experimentation, we show that nucleotide synthesis can determine the relationship between [ATP] and growth and thus the strength of inoculum effect in an antibiotic class-dependent manner. If the [ATP]/growth rate is sufficiently high as determined by exogenously supplied nitrogenous bases, the inoculum effect does not occur. This is consistent for both *Escherichia coli* and *Pseudomonas aeruginosa*. Interestingly, and separate from activity through the tricarboxylic acid cycle, we find that transcriptional activity of genes involved in purine and pyrimidine synthesis can predict the strength of the inoculum effect for β-lactam and aminoglycosides antibiotics, respectively. Our work highlights the antibiotic class-specific effect of purine and pyrimidine synthesis on the severity of the inoculum effect, which may pave the way for intervention strategies to reduce the inoculum effect in the clinic.

**IMPORTANCE**  If a bacterial population can grow and reach a sufficiently high density, routine doses of antibiotics can be ineffective. This phenomenon, called the inoculum effect, has been observed for nearly all antibiotics and bacterial species. It has also been reported to result in antibiotic failure in the clinic. Understanding how to reduce the inoculum effect can make high-density infections easier to treat. Here, we show that purine and pyrimidine synthesis affect the strength of the inoculum effect; as the transcriptional activity of pyrimidine synthesis increases, the strength of the inoculum effect for aminoglycosides decreases. Conversely, as the transcriptional activity of purine synthesis increases, the strength of the inoculum effect for β-lactam antibiotics decreases. Our work highlights the importance of nucleotide synthesis in determining the strength of the inoculum effect, which may lead to the identification of new ways to treat high-density infections in the clinic.

**KEYWORDS**  antibiotic resistance, purines, pyrimidines, nucleotide synthesis, metabolism, growth rate, density, adenine, *Escherichia coli*, *Pseudomonas aeruginosa*

**Peer Reviewers** Peter Belenky, Brown University, Providence, Rhode Island, USA; Jinki Yeom, Seoul National University College of Medicine, Seoul, South Korea

Address correspondence to Robert P. Smith, rsmith@nova.edu.

The authors declare no conflict of interest.

See the funding table on p. 20.

Antibiotic resistance poses a significant threat to global public health (1). With the antibiotic pipeline largely drying up (2), and recent projections indicating that upward of 10 million annual deaths from antibiotic resistance will occur by the year 2050 (3), we must understand the mechanisms by which bacteria tolerate and resist antibiotic treatment. While the majority of previous work has focused on how a bacterium resists

antibiotics (4), there is a growing appreciation that bacteria can resist antibiotics as a collective. For example, collective degradation of β-lactams antibiotics by β-lactamases can enhance the resistance of the entire population (5), facilitate the growth of non-resistant bacteria (6), and enhance horizontal gene transfer (7). Bacteria can make shared use of secreted extrapolymeric substances during biofilm formation, which increases antibiotic tolerance (8). The collective swarming action of bacteria can also increase antibiotic tolerance (9). Finally, bacteria can use altruistic cell death to enhance antibiotic resistance (10). Overall, understanding the mechanisms by which bacteria tolerate and resist antibiotics as a collective will lead to the development of novel ways to disrupt such cooperative behaviors, which will prolong the use of our existing antibiotics.

It has been well-established that bacteria can resist and tolerate antibiotics as a collective by increasing their population density (11). For a given concentration of antibiotic, if the density of the bacterial population is sufficiently high, the bacteria will tolerate the antibiotic and grow. Otherwise, if the density of the bacterial population is sufficiently low, the bacteria are susceptible to the antibiotic and die. This phenomenon, called the inoculum effect (IE), has been observed for multiple antibiotics and bacterial species (11–20) and can occur in the absence of canonical antibiotic resistance mechanisms (15). IE has been shown to reduce antibiotic efficacy in animal models (13, 21) and has been suspected in antibiotic treatment failure in the clinic (22–31). Moreover, antibiotic tolerance owing to IE has been postulated to drive the evolution of additional resistance mechanisms (32, 33), which further renders antibiotics ineffective. Accordingly, it is critical to understand the mechanisms that allow IE to arise to increase the efficacy and prolong the usefulness of existing antibiotics.

Multiple mechanisms to explain IE have been proposed. Some proposed mechanisms are antibiotic-specific. These include collective degradation of β-lactam antibiotics by β-lactamase producing bacteria (34, 35), degradation of ribosomes following treatment with aminoglycosides (15), and differential growth rates during quinolone treatment (18). Other proposed mechanisms are more general and cover multiple antibiotics. These include a decrease in the ratio of antibiotic to antibiotic target (35) and differences in the length of lag phase owing to changes in initial density (36). More recently, we showed that interactions between bacterial metabolism and growth rate can determine the strength of IE for bactericidal antibiotics (37). Specifically, we found that, for a given growth environment, the strength of IE is dependent on the change in adenosine triphosphate [ATP] relative to the change in growth rate, a relationship that we call growth productivity. Increasing growth productivity reduced the strength of IE; if growth productivity was sufficiently high, IE was abolished. Our previously proposed mechanism relies on the interactions between ATP, growth rate, and initial density to explain IE. Bacterial populations initiated from high density have a short period of log phase growth where [ATP] is greatest before entering the stationary phase, where ATP synthesis slows. As antibiotic lethality is dependent upon both growth rate (38) and bacterial metabolism (39), the time over which both are high is relatively short for a high-density population. Thus, they are inherently more tolerant of antibiotics. Conversely, populations initiated from lower density spend considerably more time in log phase, where both growth rate and metabolism are high and are thus more susceptible to antibiotics. This difference in antibiotic susceptibility between high and low initial-density populations accounts for IE.

Recent work has highlighted the importance of identifying the cellular pathways involved in determining antibiotic efficacy. For example, the SOS response pathway (40) and the stringent response (41) have both been implicated in affecting antibiotic resistance and tolerance. Multiple studies have implicated metabolic activity through the tricarboxylic acid (TCA) cycle as a determining factor in antibiotic lethality (42–44). More recently, it has been suggested that adenine limitation as determined through nucleotide synthesis pathways during antibiotic treatment can potentiate antibiotic efficacy (45). The identification and subsequent study of these pathways have led to the discovery of novel genes and pathways under selection during antibiotic treatment

(46) and have spurred interest in formulating novel antibiotic adjuvants (47). However, it remains unclear as to which pathway(s) are involved in determining the relationship between [ATP] and growth rate in the context of IE. Accordingly, we sought to identify this pathway, the discovery of which may lead to the identification of novel drug targets or new approaches to treating recalcitrant high-density infections.

## RESULTS

### Flux balance analysis coupled with Optknock identifies the superpathway of histidine, purine, and pyrimidine biosynthesis as a determinant of the inoculum effect

Our previous work demonstrated that flux balance analysis (FBA) could predict changes in [ATP] and growth rate (Fig. S1) as determined by the chemical composition of the growth environment. Our work also demonstrated that FBA could predict the qualitative trends between the strength of IE and the relationship between ATP and growth rate. Computationally, this was determined by computing the ratio of ATP synthesis (ATPsyn) to steady-state biomass production, which served as both the objective function and a measure of growth (37). To identify pathways involved in determining ATP production and growth rate in *Escherichia coli*, we performed FBA with Optknock (48). Optknock individually removes genes from a genomic model by eliminating the lower bound flux value. This approach is the computational equivalent of screening a library of knockout strains but has the additional benefit of being able to assess the effect of removing both non-essential and essential genes. Using a set of parameters that capture the composition of M9 medium (Table S1), we sequentially removed all 1,516 genes from the *E. coli* whole genome model. Next, we determined how each gene deletion impacted ATPsyn/biomass, a metric that approximates growth productivity. We experimentally confirmed that this approach could predict qualitative changes in experimentally determined [ATP] and growth rate by measuring both in select knockout strains (Fig. S1). Considering only genes whose removal increased ATPsyn/biomass above wild-type levels, we assigned pathways to each gene and calculated the frequency at which each pathway occurred in the data set. We found that the superpathway of histidine, purine, and pyrimidine biosynthesis occurred with the greatest frequency [Fig. 1A; fold enrichment = 3.7 and enrichment false discovery rate (FDR) <0.0001]. Several additional pathways involved in nucleotide synthesis were also found in this analysis including the superpathway of purine *de novo* biosynthesis II (fold enrichment = 4.9 and enrichment FDR < 0.0001), the superpathway of pyrimidine deoxyribonucleotides *de novo* biosynthesis (fold enrichment = 3.5 and enrichment FDR = 0.005), and the superpathway of pyrimidine ribonucleotide *de novo* biosynthesis (fold enrichment = 4.6 and enrichment FDR = 0.0045).

We then averaged ATPsyn/biomass across all genes assigned to a given pathway. For example, the independent removal of 17 different genes from the superpathway of histidine, purine, and pyrimidine biosynthesis was predicted to increase ATPsyn/biomass relative to wild type. In our analysis, we averaged the ATPsyn/biomass values for these 17 genes and compared this averaged value to all other averaged ATPsyn/biomass values assigned to other pathways. While the greatest average ATPsyn/biomass value belonged to genes in the folate polyglutamylation pathway, we found that genes in the superpathway of histidine, purine, and pyrimidine biosynthesis, the superpathway of pyrimidine deoxyribonucleotides *de novo* biosynthesis, and the superpathway of pyrimidine deoxyribonucleotides *de novo* biosynthesis (*E. coli*) were all represented in the top 20. Together, this computational analysis implicated purine and pyrimidine biosynthetic pathways in influencing ATPsyn/biomass, which we explored further.

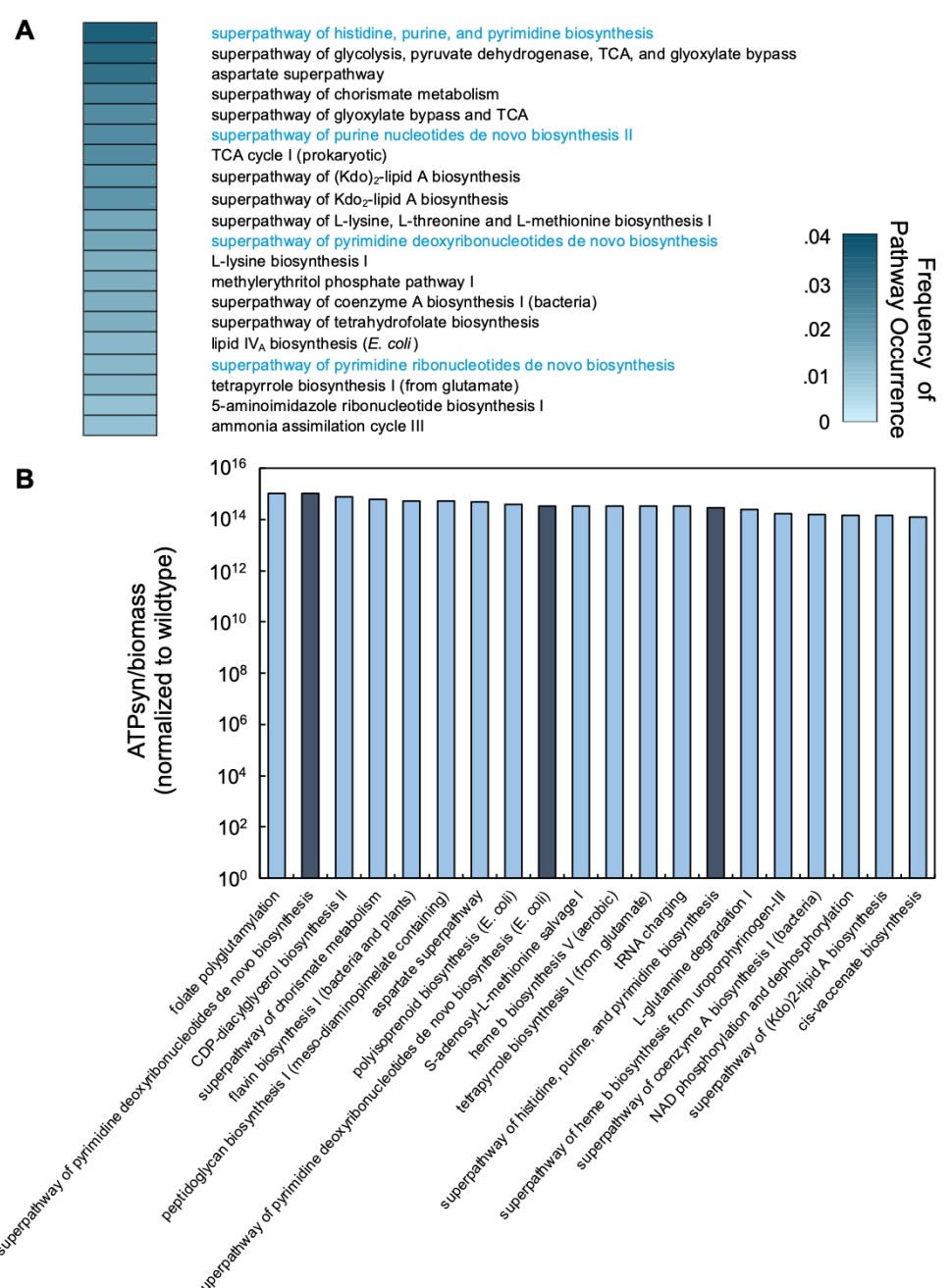

FIG 1  FBA with Optknock implicates pathways involved in nucleotide synthesis in having the most frequent effect on ATPsyn and biomass. (A) The frequency of pathways in which genes removed by OptKnock increases the ratio of ATPsyn/biomass above wild type. Each gene removed was assigned to a corresponding pathway as determined using EcoCyc. The percentage of the top 20 pathways is shown. Blue text indicates pathways involved in nucleotide synthesis. Parameters for FBA in Table S1. (B) Average ATPsyn/biomass values for the top 20 pathways with the greatest ATPsyn/biomass values. All ATPsyn/biomass values were normalized to wild type. Dark blue bars indicate pathways involved in nucleotide synthesis.

## Supplementation with nitrogenous bases alters [ATP], growth rate, and log[ATP]/growth rate

To start, we considered a minimal network that describes the synthesis of both nucleotide classes and salvage of nitrogenous bases (Fig. 2A). Nucleotide synthesis occurs as two separate pathways, purine and pyrimidine synthesis. Pyrimidine synthesis begins with glutamate, which, after several enzymatic reactions, is converted into uridine

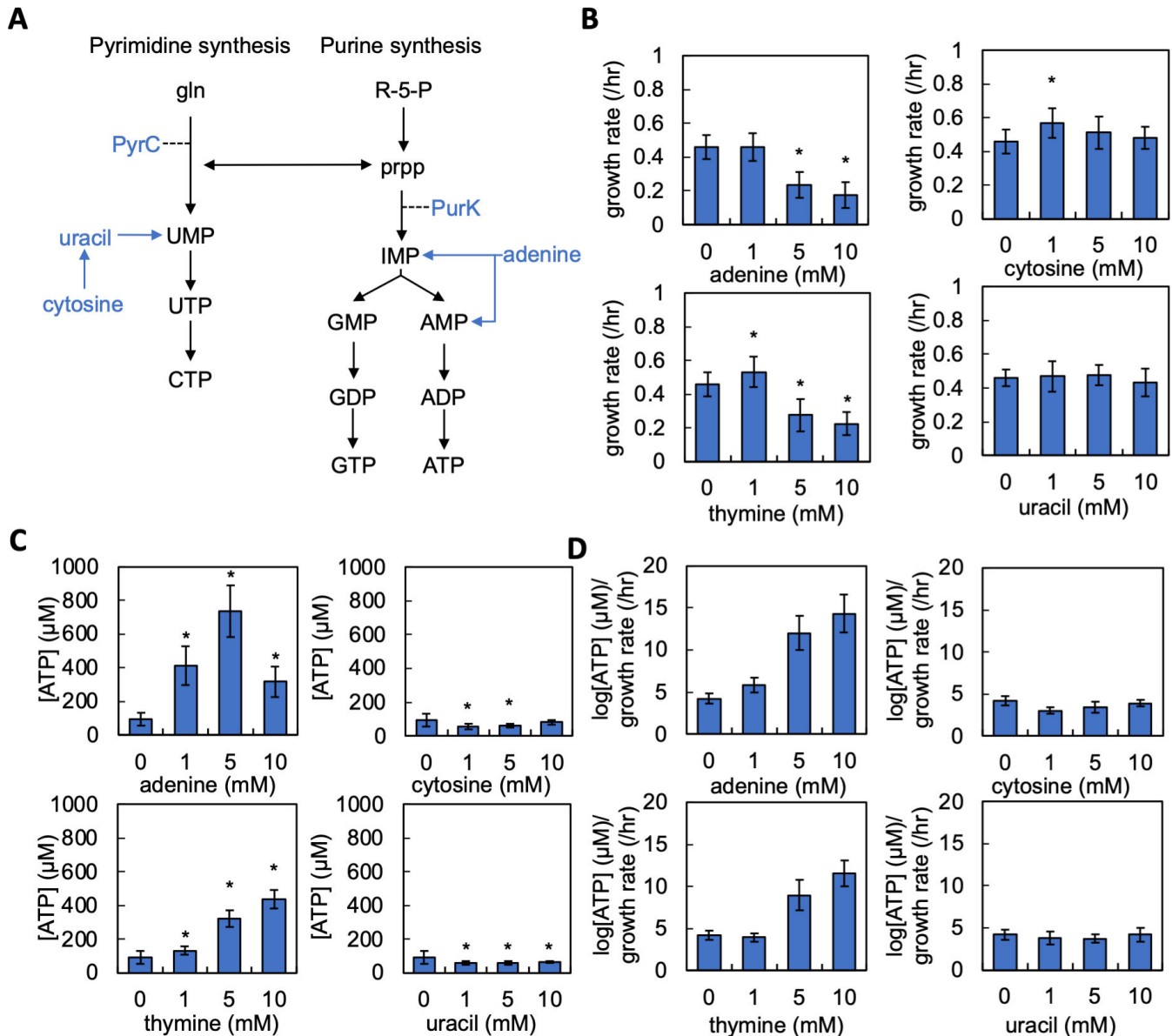

**FIG 2** Supplementing the growth medium with nitrogenous bases alters growth rate, [ATP], and the ratio of log[ATP]/growth rate. (A) A minimal network of pyrimidine and purine synthesis. gln = glutamate. R-5-P = ribose-5-phosphate. prpp = 5-phospho-α-D-ribose 1-diphosphate. PyrC and PurK are shown to facilitate understanding of data presented in Fig. 5. (B) Average growth rate of *E. coli* grown in M9 medium supplemented with adenine (top left), cytosine (top right), thymine (bottom left), or uracil (bottom right). Average plotted from five different percentages of casamino acids, each consisting of ≥4 biological replicates. Error bars =SEM. * = different than control [0 mM nitrogenous bases, *P* < 0.005 (two-tailed *t*-test); all *P* values for this figure in Table S2]. Residual values for growth curve fitting in Table S3. Growth curves in Fig. S2. Raw data in Fig. S3. (C) Average [ATP] of *E. coli* grown in M9 medium supplemented with adenine (top left), cytosine (top right), thymine (bottom left), or uracil (bottom right). Average plotted from five different percentages of casamino acids, each consisting of four biological replicates. Raw data in Fig. S4. Errors bars = SEM. * = different than control [0 mM nitrogenous bases, *P* < 0.005 (two-tailed *t*-test)]. (D) Log[ATP]/growth rate for *E. coli* grown in M9 medium supplemented with adenine (top left), cytosine (top right), thymine (bottom left), or uracil (bottom right). Data from panels B and C. Errors bars = SEM.

triphosphate (UTP). UTP is used to create cytidine triphosphate (CTP). Purine synthesis begins with ribose-5-phosphate (R-5-P), which is converted to inosine monophosphate (IMP). IMP serves as the substrate for the stepwise synthesis of both guanosine triphosphate (GTP) and ATP. 5-phospho-α-D-ribose 1-diphosphate (prpp) serves as a link between *de novo* purine and pyrimidine synthesis as it is used as a substrate for both pathways, thus allowing both pathways to autoregulate (49). A reduction

in activity in one pathway (e.g., purine synthesis) leads to the accumulation of prpp, which can subsequently be used by the other pathway (e.g., pyrimidine synthesis), thus increasing its activity (45). Nitrogenous bases can be salvaged from the surrounding environment, can be used for nucleotide synthesis, and have been shown to alter purine and pyrimidine synthesis (45, 50). Imported adenine is converted to either adenosine monophosphate (AMP) or IMP, both of which are involved in purine synthesis (51). Cytosine is imported and converted into uracil; both salvaged uracil and cytosine-derived uracil are converted to uridine monophosphate (UMP) via pyrimidine nucleo-bases salvage pathways (51). Previous work has also shown that altering nucleotide synthesis can alter bacterial metabolism, including [ATP], growth rate, and antibiotic lethality (45, 50), all of which are important in determining IE (37). Accordingly, we chose to perturb nucleotide synthesis, growth rate, and [ATP] by providing nitrogenous bases in the growth medium. Additional details on both nucleotide synthesis and salvage pathways can be found in the Supplemental Results.

To quantify the effect of exogenously supplied nitrogenous bases on growth rate and ATP production, we grew *E. coli* in M9 medium using different concentrations of cytosine, adenine, thymine, and uracil that were previously shown to be sufficient to perturb nucleotide synthesis (45, 52). Guanine could not be used as it could not be dissolved at 1 mM in M9 medium. Cultures that lacked the addition of nitrogenous bases in the growth medium served as the control. Consistent with our previous work (37), we provided different percentages of casamino acids in the M9 medium, which served as a nitrogen source. Varying casamino acids allowed us to study IE over a range of [ATP] and growth rates for a given growth environment, which was defined by the concentration and type of nitrogenous base provided.

To quantify the maximum growth rate, we performed high-resolution measurements of bacterial density ($OD_{600}$) for 10 hours in a microplate reader (37). We then fit the resulting growth curves to a logistic equation whereupon we extracted the maximum growth rate. We found that the addition of 5 or 10 mM of adenine and thymine significantly reduced the growth rate as compared to the control (Fig. 2B). The addition of 1 mM cytosine or 1 mM thymine increased growth rate relative to the control. Otherwise, no other significant changes in growth rate were noted. Importantly, the maximum growth rate does not change as a function of initial densities measured in this study (Fig. S3). Next, we quantified the effect of nitrogenous base supplementation on [ATP]. We measured [ATP] normalized by cell density using an approach that allows measurement of bacterial metabolism separate from growth (37, 39). Measurement of [ATP] under this condition is strongly correlated to other measures of metabolism, including [NAD$^+$]/[NADH] and oxygen consumption rate (37, 39). The addition of either adenine or thymine at all concentrations tested significantly increased [ATP] relative to the control. With adenine supplementation, [ATP] appeared biphasic, where the greatest levels observed occurred with 5 mM adenine (Fig. 2C). With thymine supplementation, we observed that as the concentration of thymine increased, [ATP] also increased. When the medium was supplemented with either cytosine or uracil, a significant decrease in [ATP] was observed at 1 and 5 mM. A decrease in [ATP] was also observed with 10 mM uracil.

Finally, we determined the ratio of log[ATP]/growth rate, which served as the experimental measure of ATPsyn/biomass. We found that when either uracil or cytosine was supplied in the growth medium, log[ATP]/growth rate was largely consistent with the control (Fig. 2D). However, with the addition of either 5 or 10 mM adenine and thymine, log[ATP]/growth rate was greater than the control by two- to threefold. A small increase in this ratio was also noted when 1 mM adenine was supplied in the growth medium. Overall, supplementation of adenine and thymine, but not cytosine or uracil, impacted log[ATP]/growth rate.

## Nitrogenous base-driven increases in [ATP]/growth rate decrease the strength of the inoculum effect

To assess the impact that manipulating log[ATP]/growth rate through nucleotide synthesis had on IE, we measured the minimum inhibitory concentration (MIC) of the well-studied aminoglycoside streptomycin in *E. coli*. MICs were measured for two populations initiated from high and low density. Importantly, the high-density populations ($2.03 \times 10^6$ CFU/mL $\pm$ $4.75 \times 10^5$) were above the standard density used to measure MICs in the clinic ($5.5 \times 10^5$ CFU/mL), whereas the low-density population ($1.67 \times 10^5$ CFU/mL $\pm$ $5.42 \times 10^4$) was below this density. MICs were assessed in increasing concentrations of streptomycin in M9 medium with and without nitrogenous bases (control). We could not achieve reliable growth at all percentages of casamino acids when 10 mM adenine was included in the medium. MICs in this condition could not be quantified. In general, we found that as the percentage of casamino acids increased in the growth medium, the MIC of both high- and low-density populations increased (Fig. 3A). Consistent with previous work (37), we calculated the strength of the inoculum effect ($\Delta$MIC) by determining the average difference in MIC between the high- and low-density populations. We found that when either adenine or thymine was included in the growth medium, $\Delta$MIC decreased as compared to the control (Fig. 3B); the reduction in $\Delta$MIC was significant when 5 mM adenine, 5 mM thymine, or 10 mM thymine was included in the growth medium. Interestingly, supplementation with 10 mM thymine yielded a $\Delta$MIC value that was not statistically different than zero, indicating that IE was abolished. Finally, supplementation with 1 mM adenine, 1 mM thymine, cytosine, or uracil did not significantly change $\Delta$MIC. In general, the $\Delta$MIC values observed herein are within the order of magnitude of previously observed values found with clinical isolates of *E. coli* (53).

Next, we performed a regression analysis between $\Delta$MIC and log[ATP]/growth rate. We found a strong and significant linear relationship between $\Delta$MIC of streptomycin and log[ATP]/growth rate; as log[ATP]/growth rate increased, $\Delta$MIC decreased (Fig. 3C). To test the generality of the relationship, we challenged *E. coli* with two additional antibiotics, carbenicillin ($\beta$-lactams) and ciprofloxacin (fluoroquinolones). These antibiotics are commonly used to investigate mechanisms of resistance (45, 46). Consistent with streptomycin, we found a strong and significant relationship between the $\Delta$MIC of carbenicillin and log[ATP]/growth rate (Fig. 3D). We did not find a strong or significant relationship with $\Delta$MIC of ciprofloxacin (Fig. 3E). The relationship between log[ATP]/growth rate and $\Delta$MIC of streptomycin and carbenicillin holds when a higher initial density ($2.00 \times 10^7$ CFU/mL $\pm$ $4.50 \times 10^6$) of *E. coli* is used (Fig. S8). Overall, we found that supplementation with nitrogenous bases could alter log[ATP]/growth rate, which determined $\Delta$MIC for streptomycin and carbenicillin but not ciprofloxacin.

To test alternative hypotheses that could explain the effects of nitrogenous bases on log[ATP]/growth and its relationship with $\Delta$MIC, we found that concentrations of adenine (1 mM) and thymine (0.9 mM) that altered either growth rate or [ATP], but not both, did not change $\Delta$MIC of streptomycin (Fig. S9). Importantly, these nitrogenous bases reduced $\Delta$MIC when provided in higher concentrations. We next performed regression analysis between $\Delta$MIC for each antibiotic and log[ATP] or growth rate (Fig. S9). Similar to our previous work, we found significant relationships between [ATP] and growth rate when independently correlated with $\Delta$MIC (37). However, in general, the $R^2$ values of these relationships were most often less than that of $\Delta$MIC plotted as a function of log[ATP]/growth. Our findings cannot be explained by other forms of density-dependent antibiotic resistance including quorum sensing (*luxS*) or efflux pumps (*mdtA*) as knockout strains lacking these activities continued to show IE. Moreover, supplementation with thymine, but not cytosine, reduced $\Delta$MIC in these knockout strains (Fig. S10). While previous work (15, 37) has suggested that changes to the ratio of antibiotic to antibiotic target can explain IE, we did not find a strong or significant relationship between $\Delta$MIC for streptomycin and the concentration of rRNA, which serves as a measure of ribosome concentration (Fig. S10; Supplemental Methods and Results). As ribosomes are the target

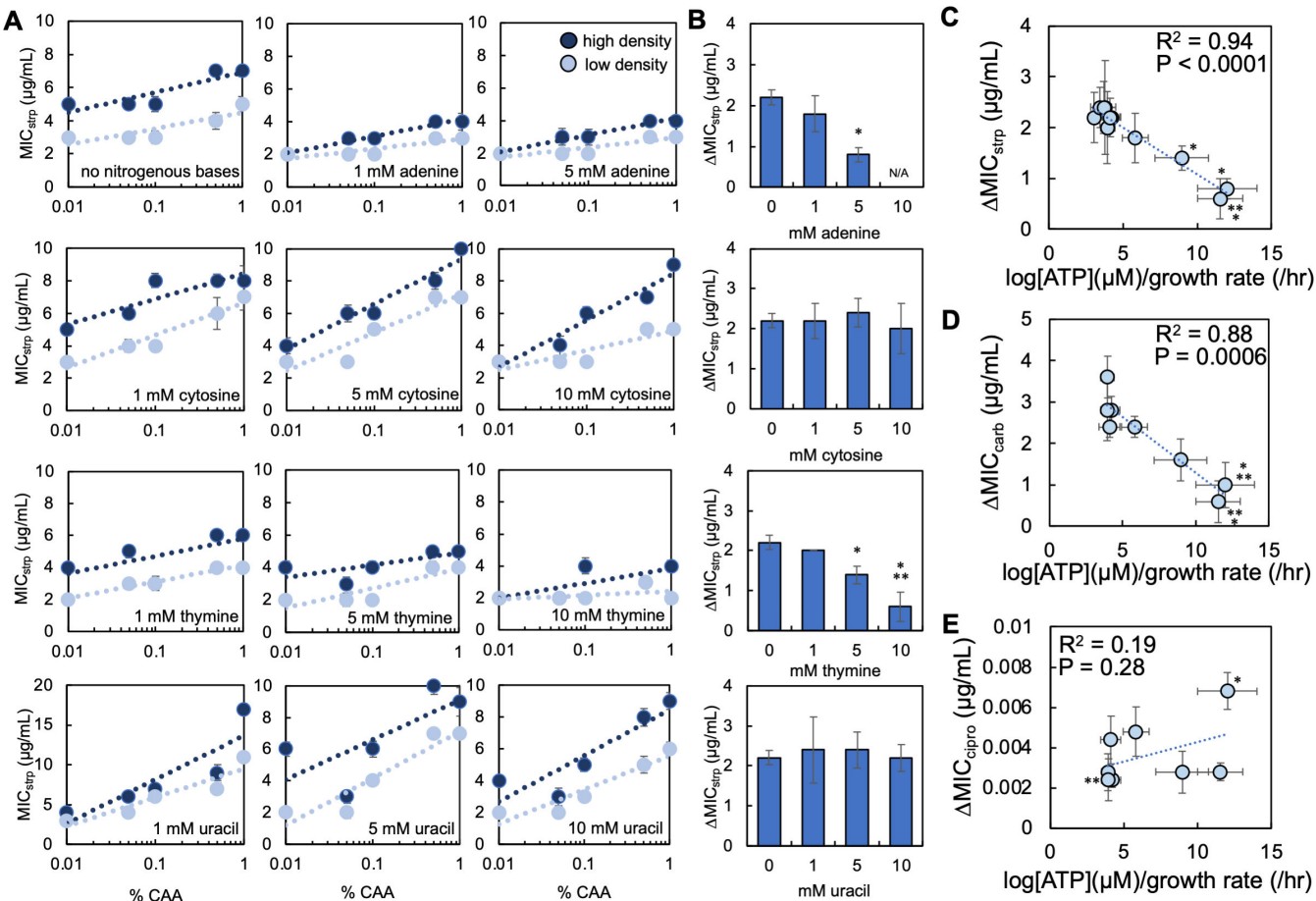

**FIG 3** Ratio of log[ATP]/growth rate predicts the strength of inoculum effect (ΔMIC) for streptomycin and carbenicillin but not ciprofloxacin. (A) MIC of initial high (dark blue) and low (light blue) density *E. coli* populations grown in M9 medium supplemented with different nitrogenous bases and with streptomycin (strp). SEM from ≥5 biological replicates. Raw data in Fig. S5. (B) Average ΔMIC for *E. coli* populations grown in M9 medium as in panel A. * Different than no nitrogenous base control ($P ≤ 0.036$, two-tailed *t*-test); ** Not different than zero ($P = 0.104$, one-tailed *t*-test). Error bars = SEM. (C) ΔMIC of strp as a function of log[ATP]/growth rate. $R^2$ and *P* value from a linear regression. Weighted least squares (WLS) regression ($R^2 = 0.98$, $P < 0.0001$); Deming regression ($P < 0.0001$). Error bars = SEM. MIC data from panel A; log[ATP]/growth from Fig. 2D. (D) ΔMIC of carbenicillin (carb) as a function of log[ATP]/growth rate. $R^2$ and *P* value from a linear regression. WLS regression ($R^2 = 0.82$, $P = 0.002$); Deming regression ($P = 0.0006$). Error bars = SEM. * Different than no nitrogenous base control ($P ≤ 0.03$, two-tailed *t*-test); ** not different than zero ($P ≥ 0.071$, one-tailed *t*-test). MIC data from Fig. S6; log[ATP]/growth from Fig. 2D. (E) ΔMIC of ciprofloxacin (cipro) as a function of log[ATP]/growth rate. $R^2$ and *P* value from a linear regression. WLS regression ($R^2 = 0.15$, $P = 0.34$); Deming regression ($P = 0.28$). * Different than no nitrogenous base control ($P = 0.009$, two-tailed *t*-test); ** not different than zero ($P = 0.054$, one-tailed *t*-test, 1 mM thymine within data cluster). Error bars = SEM. MIC data from Fig. S7; log[ATP]/growth from Fig. 2D.

of streptomycin, it is unlikely that a change in the number of antibiotic targets (ribosomes) can account for IE in our system. For a more detailed explanation of alternative hypotheses, see the Supplemental Results.

## Exogenous nitrogenous bases supplementation affects the strength of the inoculum effect in multiple bacterial species

To test the ability of nitrogenous base supplementation to alter log[ATP]/growth rate and ΔMIC in additional bacterial species, we grew *Pseudomonas aeruginosa* in M9 medium supplemented with nitrogenous bases and measured growth rate and [ATP] as above. We found that supplementation with nitrogenous bases could insignificantly alter growth rate compared to the control (Fig. 4A). Supplementation with nitrogenous bases could also alter ATP production. However, only supplementation with 5 mM adenine significantly reduced [ATP] relative to the control (Fig. 4B). Together, these

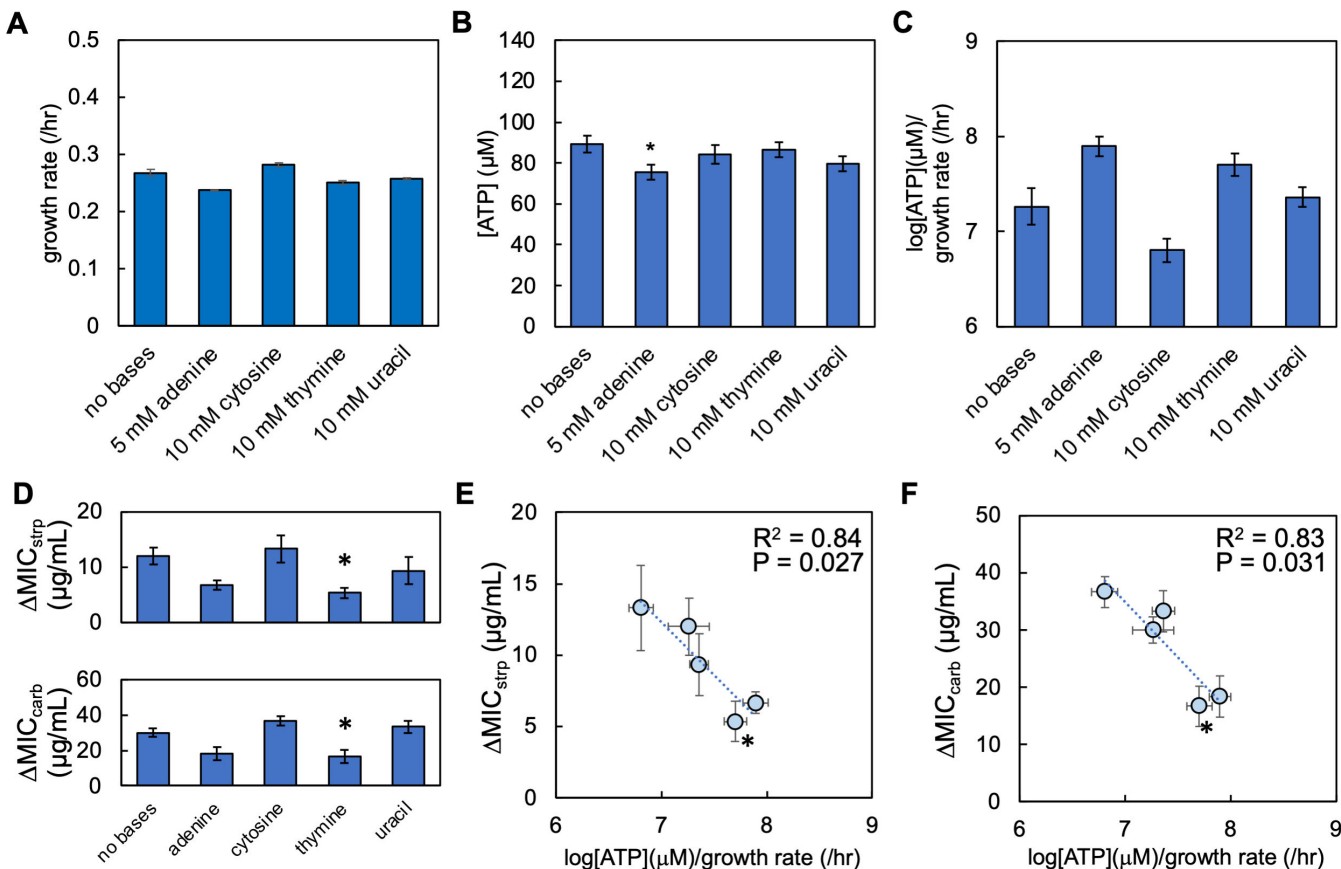

**FIG 4** Nitrogenous base-driven changes in [ATP]/growth rate determine ΔMIC in *P. aeruginosa*. (A) Average growth rate of *P. aeruginosa* grown in M9 medium supplemented with nitrogenous bases at the concentration indicated. Data from three percentages of casamino acids (0.1%, 0.5%, and 1%), each consisting of ≥3 biological replicates. Errors bars = SEM. Growth curves and raw data in Fig. S11. Residual values for growth curve fitting in Table S4. All *P* values for the figure in Table S5. (B) Average [ATP] of *P. aeruginosa* grown in M9 medium supplemented with nitrogenous bases. Average from three percentages of casamino acids, each consisting of three biological replicates. Raw data in Fig. S11. Errors bars = SEM. * = different than the no nitrogenous base control (*P* = 0.032, two-tailed *t*-test). (C) log[ATP]/growth rate for *P. aeruginosa*. Data from panels A and B. Errors bars = SEM. (D) Average ΔMIC for *P. aeruginosa* populations grown in M9 medium. Top: ΔMIC of streptomycin (strp). Bottom: ΔMIC of carbenicillin (carb). Error bars = SEM. Average from ≥4 biological replicates. Raw data in Fig. S11. * = different than the no nitrogenous base control (*P* ≤ 0.041, one-tailed *t*-test). Percentage of casamino acids and concentration of nitrogenous bases as in panel A. (E) ΔMIC of strp as a function of log[ATP]/growth rate. $R^2$ and *P* value from a linear regression. Error bars = SEM. MIC data from panel D; log[ATP]/growth from panel C. WLS: $R^2$ = 0.71, *P* = 0.073. Deming regression: *P* = 0.0272. (F) ΔMIC of carb as a function of log[ATP]/growth rate. $R^2$ and *P* value from a linear regression. Error bars = SEM. MIC data from panel D; log[ATP]/growth from panel C. WLS: $R^2$ = 0.85, *P* = 0.026. Deming regression: *P* = 0.031.

trends altered log[ATP]/growth rate, and consistent with *E. coli*, the two greatest values were observed when either 10 mM thymine or 5 mM adenine was supplemented in the medium (Fig. 4C). Next, we determined ΔMIC for streptomycin and carbenicillin. We found that supplementation with either 5 mM adenine or 10 mM thymine reduced the ΔMIC of streptomycin and carbenicillin. However, only the reduction in ΔMIC for thymine was significant relative to the control (Fig. 4D). Similar to our findings in *E. coli*, we found a strong and significant relationship between ΔMIC for both antibiotics and log[ATP]/growth rate; as log[ATP]/growth rate increased, ΔMIC decreased (Fig. 4E and F). However, when either log[ATP] or growth rate alone was plotted as a function of ΔMIC, the relationship was not consistently strong or significant (Fig. S11). The values of ΔMIC observed herein for *P. aeruginosa* are near those reported for clinical strains given the same initial starting densities (54). Overall, these findings were consistent with *E. coli*, suggesting that the relationship between log[ATP]/growth rate and ΔMIC as determined by nitrogenous base supplementation can be found in additional Gram-negatives.

## Knockout strain analysis reveals opposing effects of purine and pyrimidine synthesis on the strength of the inoculum effect

To provide direct support for the role of purine and pyrimidine synthesis in determining ΔMIC, we used FBA with OptKnock to identify genes involved in nucleotide synthesis whose removal would alter ATPsyn and biomass; together, this would alter log[ATP]/growth rate. Accordingly, we performed OptKnock simulations using parameters that captured the composition of the M9 medium without nitrogenous base supplementation. While our FBA and OptKnock analysis identified multiple genes whose removal would increase ATP/growth rate, we focused on two genes, *purK* and *pyrC*, both of which lack mammalian homologs and are involved in purine and pyrimidine synthesis, respectively (55) (Fig. 5A, see Fig. 2A for position in a network). Our computational analysis predicts that the removal of *pyrC* would increase ATP synthesis (ATPsyn) while reducing growth rate (biomass) relative to wild type (Fig. 5B and C). Conversely, the removal of *purK* would decrease both ATP synthesis and growth rate relative to wild type (Fig. 5B and C). To test these predictions, we measured growth rate and [ATP] as above. While the growth of *E. coli* lacking these genes was achieved in LB medium, growth was not consistently observed in traditional M9 medium (Fig. S12). This was consistent with FBA and Optknock predictions that predicted biomass values of 0 when these genes were removed. However, the growth of these strains could be restored in M9 medium if low equimolar concentrations (1, 4, and 7 µM) of all five nitrogenous bases were provided. Consistent with our model predictions, we found that removal of *pyrC* (Δ*pyrC*) resulted in elevated [ATP]. Conversely, [ATP] in a strain lacking *purK* (Δ*purK*) was lower than the wild type (Fig. 5D). We also found that both Δ*pyrC* and Δ*purK* had reduced growth rates relative to the wild type (Fig. 5E). As predicted computationally, these changes led to greater log[ATP]/growth rate ratios as compared to wild type (Fig. 5F).

We then quantified the ΔMIC of streptomycin and carbenicillin for both strains in addition to the wild-type strain. ΔMIC of streptomycin and carbenicillin decreased significantly and was not different than zero for Δ*pyrC* relative to the wild type (Fig. 5G and H). Conversely, for Δ*purK*, ΔMIC was no different than wild type when challenged with streptomycin. However, when challenged with carbenicillin, ΔMIC increased relative to the wild type. Interestingly, these trends in ΔMIC did not follow trends in log[ATP]/growth rate. While the increase in log[ATP]/growth rate in Δ*pyrC* was consistent with the reduction in ΔMIC for both antibiotics, the increase in log[ATP]/growth in Δ*purK* did not alter (streptomycin) or increased (carbenicillin) ΔMIC. These data suggest that genetic inhibition of purine synthesis (Δ*purK*) protects bacteria by increasing (carbenicillin), or not altering (streptomycin), ΔMIC. On the other hand, genetic inhibition of pyrimidine synthesis (Δ*pyrC*) decreases ΔMIC, effectively abolishing IE for both antibiotics and is consistent with the observed increase in log[ATP]/growth rate.

## Chemical perturbation of purine and pyrimidine synthesis pathways alters ΔMIC

Our genetic analysis implied non-equivalent effects of purine and pyrimidine synthesis in determining ΔMIC. It also implied that the relationship between ΔMIC and log[ATP]/growth could be disrupted through direct genetic perturbation of both pathways. However, the removal of key enzymes in both nucleotide synthesis pathways would alter regulation in and between both pathways by removing key regulatory steps, which could also alter the energetic requirements of both pathways. In addition to autoregulation (50), negative feedback plays a key role in regulating activity within each pathway (Fig. 6A). During purine synthesis, accumulation of AMP represses PurF, which attenuates purine synthesis (56). However, accumulation of IMP, a purine synthesis intermediate from which AMP is produced, potentiates the activity of pyrimidine synthesis. UTP produced by pyrimidine synthesis attenuates the activity of several upstream enzymes, including PyrH (57) and PyrE (58). UMP also inhibits the first steps of pyrimidine synthesis (59). To preserve the regulation both within and between both nucleotide synthesis

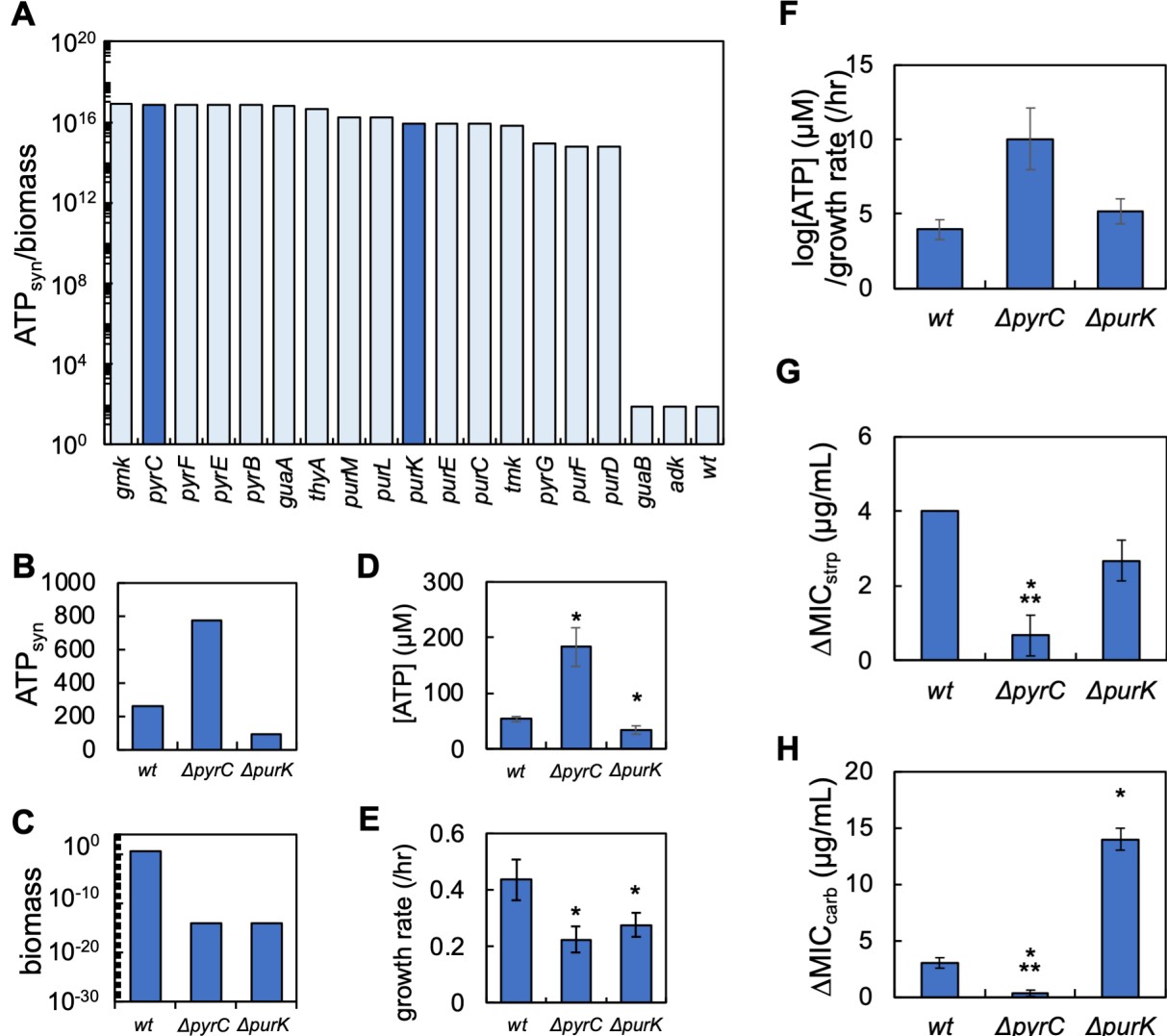

**FIG 5** Knockout strains reveal antibiotic-specific effects of purine and pyrimidine synthesis on ΔMIC for streptomycin and carbenicillin. (A) Simulation: genes identified by FBA and OptKnock that are involved in nucleotide synthesis and that increase ATPsyn/biomass above wild type (*wt*). Dark blue bars indicate *pyrC* and *purK*. FBA parameters in Table S1. Positions of PyrC and PurK in nucleotide synthesis pathway shown in Fig. 2. (B) Simulations: FBA-predicted effects on ATP synthesis (ATPsyn) for *wt*, Δ*pyrC*, and Δ*purK*. (C) Simulation: FBA-predicted effects on biomass for *wt*, Δ*pyrC*, and Δ*purK*. (D) Removal of *pyrC* increases [ATP], while removal of *purK* decreases [ATP], relative to *wt*. Error bars = SEM. Average from three biological replicates. * Indicates different than *wt* ($P < 0.01$, two-tailed *t*-test). For D and E, raw data in Fig. S12 and all $P$ values in Table S6. (E) Removal of *pyrC* and *purK* decreases growth rate relative to *wt*. Error bars = SEM. Average from ≥5 biological replicates. * Indicates different than control ($P < 0.001$, two-tailed *t*-test). Residuals for curve fitting in Table S7. (F) Removal of *pyrC* and *purK* increases log[ATP]/growth rate relative to *wt*. Data from panels D and E. (G) ΔMIC of streptomycin (strp). Error bars = SEM. Average plotted from ≥5 biological replicates. * = different than *wt* (two-tailed *t*-test, $P = 0.038$), and ** not different than zero (one-tailed *t*-test, $P = 0.211$). For G and H, raw data in Fig. S13. (H) ΔMIC of carbenicillin (carb). Error bars = SEM. Average plotted from ≥4 biological replicates. * = different than *wt* ($P < 0.025$, two-tailed *t*-test). ** Not different than zero (one-tailed *t*-test, $P = 0.211$).

pathways, we sought to chemically manipulate nucleotide synthesis, allowing greater insight into the role of purine and pyrimidine synthesis in determining ΔMIC.

To inhibit purine synthesis, we grew *E. coli* in the presence of a sub-lethal concentration of 6-mercaptopurine (6-MP), which inhibits purine synthesis, in part, by disrupting PurF activity (60–62). We focused on completing our experiments in one percentage of casamino acids, 0.1%, which was the intermediate concentration used previously (Fig. 3) and had ΔMICs that were not different than the average ΔMIC for all percentages of casamino acids measured (Fig. S14). We focused on using 5 mM adenine and 10 mM

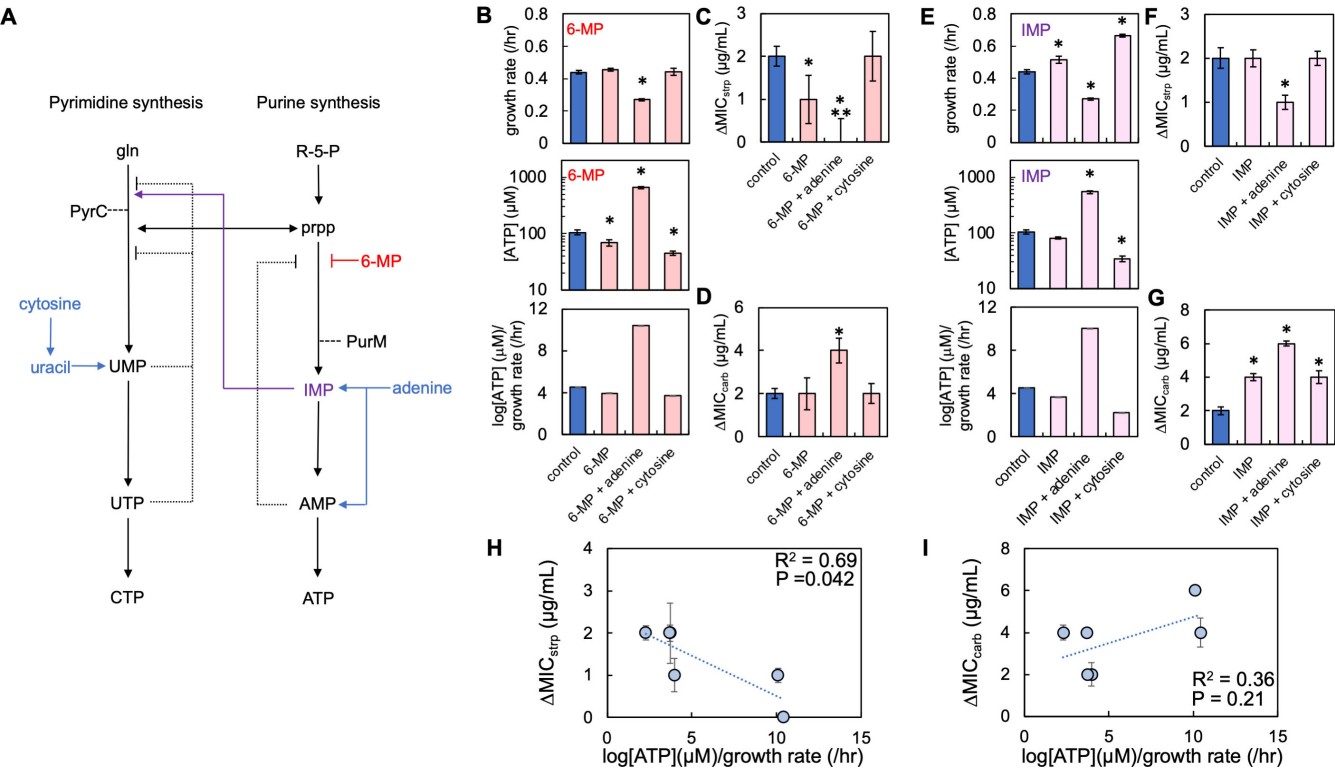

**FIG 6** Chemical manipulation of nucleotide synthesis alters ΔMIC in an antibiotic-specific manner. (A) A minimal network of pyrimidine and purine synthesis and salvage (blue) showing regulatory steps. red = 6 MP, purple = IMP, and gln = glutamate, R-5-P = ribose-5-phosphate, prpp = 5-phospho-α-D-ribose 1-diphosphate. (B) Average growth rate (top), [ATP] (middle), and log[ATP]/growth rate ratio (bottom) of *E. coli* grown in M9 medium supplemented with 5 mM adnine and 10 mM cytosine in the presence of 6-mercaptopurine (6-MP). Errors bars = SEM. * = different than control [no 6-MP, no nitrogenous bases, $P \leq 0.031$ (two-tailed *t*-test)]. For panels B–D, raw data in Fig. S15. For panels B and E, residuals for curve fitting in Table S8 (*P* values in Table S9) ΔMIC of *E. coli* grown in M9 medium supplemented with 6-MP, nitrogenous bases, and streptomycin (strp). * = difference in ΔMIC relative to control (no 6-MP, no nitrogenous bases, $P \leq 0.009$, two-tailed *t*-test). ** = not different than zero ($P = 0.31$, one-tailed *t*-test). SEM from ≥5 biological replicates. (D) ΔMIC of *E. coli* grown in M9 medium supplemented with 6-MP, nitrogenous bases, and carbenicillin (carb). * = difference in ΔMIC as compared to control (no 6-MP, no nitrogenous bases $P = 0.006$, two-tailed *t*-test). SEM from ≥5 biological replicates. (E) Average growth rate (top), average [ATP] (middle), and log[ATP]/growth rate ratio (bottom) of *E. coli* grown in M9 medium supplemented with IMP and nitrogenous bases. Errors bars = SEM. * = different than control (no IMP, no nitrogenous bases, $P \leq 0.026$, two-tailed *t*-test). For panels E–G, raw data in Fig. S16. (F) ΔMIC of *E. coli* populations grown in M9 medium supplemented with IMP, nitrogenous bases, and strp. * = difference in ΔMIC as compared to control (no IMP, no nitrogenous bases, $P = 0.009$, two-tailed *t*-test). SEM from ≥5 biological replicates. (G) ΔMIC of *E. coli* grown in M9 medium supplemented with IMP, nitrogenous bases, and carb. * = difference in ΔMIC as compared to control (no IMP, no nitrogenous bases, $P \leq 0.027$, two-tailed *t*-test). SEM from ≥5 biological replicates. (H) ΔMIC of strp as a function of log[ATP]/growth rate. $R^2$ and *P* value from a linear regression. WLS regression ($R^2 = 0.81$, $P = 0.036$). Deming regression: $P = 0.042$. Error bars = SEM. MIC data from panels C and F; log[ATP]/growth from panels B and E. (I) ΔMIC of carb as a function of log[ATP]/growth rate. $R^2$ and *P* value from a linear regression. WLS regression ($R^2 = 0.69$, $P = 0.042$). Deming regression: $P = 0.21$. Error bars = SEM. MIC data from panels D and G; log[ATP]/growth from panels B and E.

cytosine as they represented the highest concentrations of nitrogenous bases that did, or did not, affect log[ATP]/growth rate and ΔMIC (Fig. 3). Moreover, unlike thymine, salvage directly influences activity each nucleotide synthesis pathway (63). We found that 6-MP alone had no appreciable effect on growth rate but reduced [ATP] relative to the control (*E. coli* without 6-MP and nitrogenous bases); together, this resulted in a lower log[ATP]/growth rate as compared to the control (Fig. 6B). Supplementation with adenine and 6-MP reduced growth rate and increased [ATP], which increased log[ATP]/growth rate (Fig. 6B). Finally, supplementation with cytosine and 6-MP did not alter growth rate but reduced [ATP], which led to a decrease in log[ATP]/growth rate (Fig. 6B). These perturbations had largely opposed effects on ΔMIC for both antibiotics. 6-MP alone significantly reduced ΔMIC of streptomycin (Fig. 6C) but not of carbenicillin (Fig. 6D). Both 6-MP and adenine reduced ΔMIC of streptomycin such that it was not different than zero but

significantly increased ΔMIC of carbenicillin. Finally, supplementation with both 6-MP and cytosine did not result in a change in ΔMIC for both antibiotics.

Next, we grew *E. coli* in the presence of the purine synthesis pathway intermediate inosine-5-monophosphate (IMP, see Supplemental Results for an explanation on IMP import). Owing to the regulatory steps in purine synthesis, providing IMP would itself not directly repress purine synthesis; repression instead would be provided by accumulated AMP produced later in the pathway (56) (Fig. 6A). IMP also increases pyrimidine synthesis (64). Supplementation with IMP increased growth relative to the control (no IMP, no nitrogenous bases) but had no significant effect on [ATP]; together, this resulted in a small reduction in log[ATP]/growth (Fig. 6E). IMP and adenine decreased growth rate and increased [ATP], which resulted in an increase in log[ATP]/growth rate (Fig. 6E). Finally, IMP and cytosine significantly increased growth rate and reduced [ATP]; together, this reduced log[ATP]/growth rate (Fig. 6E). Consistent with our findings using 6-MP, we found that the use of IMP had opposing effects on ΔMIC for streptomycin and carbenicillin. Treatment with IMP alone left the ΔMIC of streptomycin unchanged (Fig. 6F) but significantly increased the ΔMIC of carbenicillin (Fig. 6G). Both IMP and adenine reduced the ΔMIC of streptomycin, while that of carbenicillin increased significantly. Finally, both IMP and cytosine left ΔMIC for streptomycin unchanged but increased it for carbenicillin.

Interestingly, supplementation of the growth medium with IMP or 6-MP reduced or abolished the relationship between ΔMIC and log[ATP]/growth rate and was thus generally consistent with our findings using the Δ*pyrC* and Δ*purK* knockout strains. While this relationship remained significant for streptomycin (Fig. 6H), it was weakened as compared to when only nitrogenous bases were provided in the growth medium (Fig. 3). For carbenicillin, the relationship between ΔMIC and log[ATP]/growth rate was not strong or significant (Fig. 6I). Overall, we found that chemical perturbations to nucleotide synthesis had largely opposing effects on ΔMIC for streptomycin and carbenicillin; a decrease in ΔMIC for streptomycin often led to an increase in ΔMIC for carbenicillin.

## Relative flux through purine and pyrimidine synthesis can account for opposing trends in ΔMIC

To gain insight into why manipulating purine and pyrimidine synthesis using inhibitors led to opposing effects on ΔMIC for streptomycin and carbenicillin, we used FBA to simulate the effects of the experimental perturbations above on purine synthesis and pyrimidine synthesis. To verify these FBA predictions, we used reporter strains that contained low-copy plasmids with promoters reporting activity of purine (*purM*-gfp) and pyrimidine (*pyrC*-gfp) synthesis (see Fig. 6A positions of *purM* and *pyrC* in each pathway). Importantly, pathway activity can be accurately measured using these reporter strains as many of the regulatory steps in nucleotide synthesis occur at the transcriptional level (65–69). Together, this would allow us to determine how each of the perturbations above differentially impacted activity through purine and pyrimidine synthesis.

We first simulated the effect of adding only adenine or cytosine to the growth medium. FBA predicts that adenine reduces purine synthesis activity, while pyrimidine synthesis activity increases (Fig. 7A, top panels). Conversely, the addition of cytosine left purine synthesis relatively unchanged while decreasing pyrimidine synthesis. These predictions were largely consistent with the transcriptional activity of *pyrC* and *purM* as determined by our reporter strains (Fig. 7A, bottom panels). An exception to this was transcriptional activity reported by the *purM* reporter strain which showed a significant, albeit small, reduction in transcriptional activity when cytosine was provided in the medium. This discrepancy may reflect the fact that the FBA model does not contain all reactions present in the cell and assumes that the system has reached steady state. Nevertheless, our FBA predictions largely matched the transcriptional activity of the reporter strains, and these findings were consistent with literature (45, 49).

FBA predicts that the addition of 6-MP reduces purine synthesis while having little to no effect on pyrimidine synthesis (Fig. 7B, top). The addition of both 6-MP and adenine further reduces purine synthesis concomitant with an increase in pyrimidine synthesis.

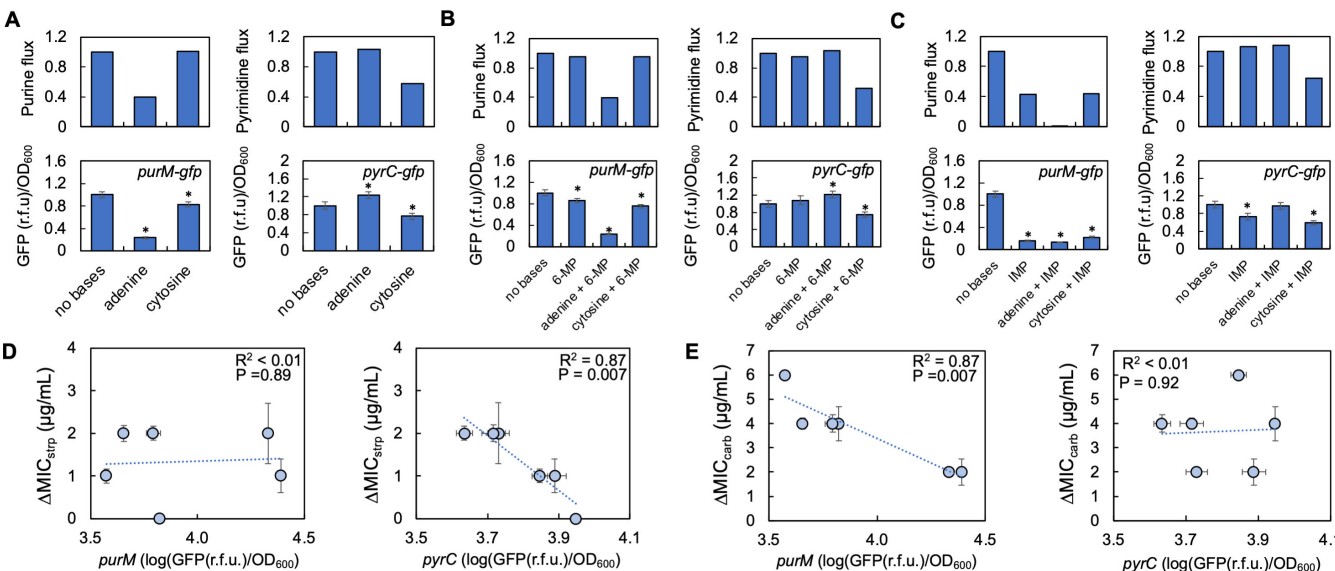

FIG 7 FBA-predicted changes in transcriptional activity of genes involved in purine and pyrimidine synthesis account for the opposing trends in ΔMIC for streptomycin and carbenicillin. (A) Top: FBA-predicted activity through purine synthesis (top left) and pyrimidine synthesis (top right) supplemented with adenine and cytosine. For panels A–C, flux activity normalized to no nitrogenous bases (no base) control. Sensitivity analysis in Fig. S17. Near identical predictions occur when multiple reactions in purine and pyrimidine synthesis are considered (Fig. S17). Bottom: transcriptional activity of *purM* (left) and *pyrC* (right) reporter strains after 18 hours of growth. r.f.u. = relative fluorescent units. For all experimental data, SEM from five biological replicates. * = different from no bases control ($P \leq 0.019$, two-tailed *t*-test). Adenine = 5 mM and cytosine = 10 mM. For panels A–C, GFP/OD$_{600}$ normalized to no base control. Transcriptional activity measured at 24 hours shows similar trends (Fig. S18). (B) Top: FBA-predicted activity through purine synthesis (left) and pyrimidine (right) synthesis supplemented with 6-MP, adenine, and cytosine. Simulation trends are consistent when 6-MP inhibits both PurF and PyrC activity (Fig. S18), which has been reported previously (70). Bottom: transcriptional activity of *purM* (left) and *pyrC* (right) reporter strains. * = $P \leq 0.035$. (C) Top: FBA-predicted activity through purine synthesis (left) and pyrimidine (right) synthesis supplemented with IMP, adenine, and cytosine. Bottom: transcriptional activity of *purM* (left) and *pyrC* (right) reporter strains. * = $P \leq$ 0.017. (D) ΔMIC of streptomycin (strp) as a function of *purM* (left) and *pyrC* (right) reporter transcriptional activity. $R^2$ and $P$ value from a linear regression. WLS regression (*purM*: $R^2 < 0.01$, $P = 0.93$; *pyrC*: $R^2 = 0.83$, $P = 0.032$); Deming regression (*purM* - $P = 0.89$; *pyrC* - $P = 0.007$). Error bars = SEM. ΔMIC from Fig. 6. For panels D and E, trends are consistent when reporter activity is measured at 24 hours (Fig. S18). (E) ΔMIC of carbenicillin (carb) as a function of *purM* (left) and *pyrC* (right) transcriptional activity (left panel). $R^2$ and $P$ value from a linear regression. WLS regression (*purM*: $R^2 = 0.85$, $P = 0.009$; *pyrC*: $R^2 = 0.32$, $P = 0.24$); Deming regression (*purM* - $P = 0.007$; *pyrC* $P = 0.89$). Error bars = SEM. MIC data from Fig. 6.

Supplementation with both 6-MP and cytosine is predicted to reduce both purine and pyrimidine synthesis. As above, these predictions were generally consistent with activity from the reporter strains (Fig. 7B, bottom). A significant reduction in *purM* promoter activity, but not *pyrC* promoter activity, was observed with 6-MP, thus confirming the inhibitory effect of 6-MP on purine synthesis (62). Supplementation with both 6-MP and adenine significantly reduced *purM* promoter activity while increasing *pyrC* promoter activity. Finally, supplementation with cytosine and 6-MP reduced both *purM* and *pyrC* reporter activity. Returning to our FBA, we found that supplementation with IMP, IMP with adenine, and IMP with cytosine results in a reduction in activity through purine synthesis (Fig. 7C, top), which was confirmed using the *purM* reporter strain (Fig. 7C, bottom). Supplementation with IMP, as well as with both IMP and adenine, was predicted to increase the activity of pyrimidine synthesis. Conversely, supplementation with both IMP and cytosine was predicted to decrease pyrimidine synthesis. While this general trend was confirmed using the *pyrC* reporter, we did observe a significant reduction in transcriptional activity in the *pyrC* promoter strain when IMP alone was provided in the growth medium. Nevertheless, our findings indicate that the above perturbation altered the transcriptional activity of genes involved in purine and pyrimidine synthesis, which were largely consistent with flux activity as determined using FBA.

To determine if differences in the transcriptional activity of purine and pyrimidine synthesis could account for the opposing changes in ΔMIC, we performed a linear regression between the transcriptional activity of purine and pyrimidine synthesis and

ΔMIC for both antibiotics. We found that while there was no significant relationship between ΔMIC for streptomycin and transcriptional activity of purine synthesis, there was a strong and significant relationship with pyrimidine transcriptional activity; as transcriptional activity of *pyrC* increased, ΔMIC of streptomycin decreased (Fig. 7D). These trends were also consistent with kanamycin, implying that this relationship is also found in additional aminoglycosides (Fig. S19). Interestingly, we found the opposite for ΔMIC of carbenicillin (Fig. 7E). Here, the relationship between the transcriptional activity of purine synthesis was strongly and significantly correlated to ΔMIC of carbenicillin; as the transcriptional activity of *purM* increased, ΔMIC of carbenicillin decreased. Conversely, there was neither a strong nor significant relationship between the transcriptional activity of pyrimidine synthesis and the ΔMIC of carbenicillin. We did neither find a significant nor strong relationship between transcriptional activity in the TCA cycle, as measured using a succinate dehydrogenase reporter, and ΔMIC of all antibiotics measured in this study (Fig. S20). This suggests that the opposing trends in ΔMIC cannot be explained by transcriptional activity in the TCA cycle. Taken together, our findings suggest that activity through pyrimidine synthesis can account for ΔMIC in representative aminoglycosides, whereas activity through purine synthesis can account for ΔMIC of representative β-lactams.

## DISCUSSION

Herein, we have provided evidence that purine and pyrimidine synthesis can impact the strength of IE. We found that log[ATP]/growth rate could predict the strength of IE for carbenicillin (β-lactam) and streptomycin (aminoglycoside) but not ciprofloxacin (fluoroquinolone). Using knockout strains and chemical modifiers of purine and pyrimidine synthesis, we showed that the strength of IE could be decoupled from log[ATP]/growth rate. Instead, our data suggest that activity through purine and pyrimidine synthesis can account for the strength of IE but is dependent upon antibiotic class. Transcriptional activity in pyrimidine synthesis predicts ΔMIC of aminoglycosides, whereas transcriptional activity in purine synthesis predicts ΔMIC in β-lactams. Taken together, our results implicate nucleotide synthesis as being a core pathway determining the strength of IE.

Recent work has highlighted the importance of nucleotide synthesis in determining antibiotic efficacy. For example, trimethoprim interferes with dihydrofolate reductase activity, negatively impacting nucleic acid synthesis and altering intracellular adenine concentrations (71, 72). The oxidation of guanine to 8-oxo-guanine during exposure to quinolones and β-lactams results in double-stranded breaks, leading to cell death (73). Adenine limitation has been proposed to increase ATP production, resulting in the production of reactive oxygen species and enhancing antibiotic lethality (45). Exogenous thymine and adenosine have also been shown to increase antibiotic lethality (52, 74). Given the multiple roles that nucleotides have in determining antibiotic lethality, alterations in their synthesis can manipulate antibiotic efficacy. Mutations in genes involved in purine synthesis were associated with increased resistance to rifampicin in *Staphylococcus aureus* (75). Mutations to both purine and pyrimidine synthesis confer resistance in *E. coli* (76). Finally, overexpression of genes that deplete cofactor pools required for nucleotide synthesis increases antibiotic tolerance (77). Our research adds to our growing understanding of the role that nucleotides and their synthesis play in determining antibiotic efficacy.

Our previous work identified that the relationship between [ATP] and growth rate, which we called growth productivity, can account for IE across multiple antibiotic classes (37). Increasing growth productivity decreased the strength of IE (37). In this study, we found that increasing log[ATP]/growth rate, a simplification of growth productivity, using nitrogenous bases reduced ΔMIC, implicating nucleotide synthesis as a determining factor in IE. How do the present findings fit within our previously proposed mechanism? As bacteria enter a stationary phase where the growth rate is significantly reduced, a reduction in DNA replication (78) and nucleotide pools (79) has been reported. Due

to the high energetic cost of producing nucleotides (80), the decrease in [ATP] owing to entry into the stationary phase may result, in part, from a reduction in nucleotide synthesis for DNA replication. Conversely, DNA synthesis and nucleotide pools are increased during the exponential phase where bacteria are most sensitive to antibiotics. As DNA and nucleotide synthesis generally correlate to the growth phase and [ATP], the role that purine and pyrimidine synthesis appear to play in determining IE fits within our previously proposed mechanism. The lack of correlation between ΔMIC of ciprofloxacin and log[ATP]/growth rate provides additional support to the role of nucleotide synthesis in determining IE (Fig. 3). Ciprofloxacin targets DNA gyrase. It causes death, in part, due to the formation of single- and double-stranded breaks during DNA synthesis (81). However, during the stationary phase, DNA gyrase is tasked with restoring DNA supercoiling (82), not DNA replication. Coupled with a reduction of [ATP] during the stationary phase, the activity of this enzyme is limited (18). Accordingly, even when perturbing [ATP], growth rate, and nucleotide synthesis, if the molecular target of the antibiotic is inactive (e.g., DNA gyrase), such perturbations are unlikely to affect antibiotic lethality. This finding is consistent with our previous work showing that the relationship between ΔMIC of ciprofloxacin can be insensitive to changes in growth productivity (37).

Initially, we found that nitrogenous base-driven changes in log[ATP]/growth rate could account for ΔMIC for streptomycin and carbenicillin (Fig. 3). However, when purine and pyrimidine synthesis were directly perturbed using gene deletion (Fig. 5) or chemical inhibitors (Fig. 6), the relationship between log[ATP]/growth and ΔMIC were significantly weakened or no longer significant. We also did not find significant relationships between ΔMIC and transcriptional activity of succinate dehydrogenase of the TCA cycle, which is often used as a measure of bacterial metabolism (45). Instead, we found that transcriptional activity of purine and pyrimidine synthesis could better explain ΔMIC for β-lactams and aminoglycosides, respectively. How can log[ATP]/growth rate account for ΔMIC when only nitrogenous bases are provided in the growth medium but not when nucleotide synthesis inhibitors are used? One possibility is that exogenous nitrogenous bases affect biochemical processes outside of bacterial metabolism and growth rate, which could impact antibiotic tolerance. For example, uracil impacts biofilm formation (83) and quorum sensing (84), both of which can influence antibiotic tolerance (85, 86), metabolism, and growth (87, 88). We also found that the removal of *pyrC* reduced ΔMIC for both carbenicillin and streptomycin (Fig. 5), which conflicts with our findings showing that pyrimidine transcriptional activity does not correlate to ΔMIC of carbenicillin (Fig. 7). One reason for this is that the removal of *pyrC* would disrupt regulatory steps in pyrimidine synthesis and crosstalk with purine synthesis, which could impact the relationship between transcriptional activity of nucleotide synthesis pathways and ΔMIC. Interestingly, the opposing effects of purine and pyrimidine synthesis on ΔMIC of β-lactam and aminoglycoside are consistent with previous work showing the effect of these pathways on antibiotic lethality. Inhibition of purine synthesis through gene deletion or the addition of chemical inhibitors decreases lethality for β-lactams while increasing it for aminoglycosides. Similarly, increasing purine synthesis through exogenously provided activators potentiates lethality for β-lactams but protects bacteria against aminoglycosides. While increased bacterial metabolism owing to adenine limitation was suggested to account for these trends (45), our work suggests that pathway activity alone can account for IE. The reason(s) for this remain unclear and should be explored in the future.

We currently lack approaches to increase the sensitivity of high-density bacterial populations to antibiotics. While increasing the concentration of antibiotics administered has shown promise in eliminating high-density infections in animal models, in humans, this may lead to off-target toxicities (89). Accordingly, targeted approaches to treat IE in the clinic are required. Our data have shown that removing *pyrC*, a gene that does not have a mammalian homolog (55), can reduce IE for both carbenicillin and streptomycin. *pyrC* encodes a dihydroorotase enzyme involved in the synthesis of pyrimidines in *E. coli* and additional pathogens. Recent work in *Acinetobacter baumanii* has demonstrated

that dihydroorotases may represent a novel class of antibiotic targets (90). Specifically, two triazolopyrimidine/imidazopyrimidine analogs initially developed as anti-malarial compounds showed *in vitro* and *in vivo* efficacy against *A. baumanii*. Similar compounds that target the *pyrC* protein may be able to reduce IE. This approach would not only increase the efficacy of existing β-lactams and aminoglycosides against high-density infections but could also reduce the evolution of antibiotic resistance, which has been associated with IE when high-density populations are treated with antibiotics at sub-MIC levels.

## MATERIALS AND METHODS

### Experimental design, strains, and growth conditions

We used *E. coli* strain BW25113 [F- Δ(*araD-araB*)*567ΔlacZ4787:rrnB*-3 λ- rph-1Δ(*rhaD-rhaB*)*568 hsdR514*] for the majority of our experiments. *P. aeruginosa* PA14 (BEI Resources) was used where indicated. *E. coli* knockout strains from the Keio collection (91) were acquired from Horizon Discovery (Boyertown, PA). Previously created reporter strains that contain plasmid-borne (SC101 origin of replication, kanamycin resistance) (92) copies of *E. coli* promoters driving the expression of *gfpmut2* were acquired from Horizon Discovery. Plasmids from these strains were isolated using a ZymoPURE plasmid miniprep kit (Zymo Research Corporation, Irvine, CA) and transformed into BW25113 using a Mix and Go Transformation kit (Zymo Research) as per the manufacturer's recommended protocol. Each experiment was initiated from an overnight culture, which was created from single colonies of bacteria grown on lysogeny broth (LB) agar medium (MP Biomedicals, Solon OH). Colonies were inoculated into 3 mL of LB liquid medium contained in 15 mL culture tubes (Genesee Scientific, Morrisville, NC). Cultures were shaken for 24 hours at 250 revolutions per minute (RPM) and 37°C, which allows bacteria to achieve stationary phase (37). Experiments were performed in modified M9 medium [1× M9 salts (48 mM $Na_2HPO_4$, 22 mM $KH_2PO_4$, 862 mM NaCl, and 19 mM $NH_4Cl$), 0.5% thiamine (Alfa Aesar, Ward Hill, MA), 2 mM $MgSO_4$, 0.1 mM $CaCl_2$, and 0.04% glucose] with various percentages of casamino acids (0.01%, 0.05%, 0.1%, 0.5%, and 1%, as indicated). Where indicated, the growth medium was supplemented with adenine (Alfa Aesar), cytosine (Acros Organics, Geel, Belgium), guanine (Thermofisher, Waltham, MA), thymine (Thermofisher), or uracil (Sigma Aldrich, St. Louis, MO) at the concentration indicated. We were unable to dissolve guanine at a final concentration of 1 mM or greater in M9 medium under the growth conditions used throughout the study. Knockout strains were grown in M9 medium supplemented with 1% casamino acids and with equimolar concentrations of all five nitrogenous bases at concentrations of 1, 4, and 7 µM. 6-MP (Alfa Aesar) and IMP (Thermofisher) were supplemented where indicated at final concentrations of 0.05 µg/mL and 1 mM, respectively.

### Flux balance analysis

Flux balance analysis was performed using the COBRA toolbox v.3.0 (93) coupled with the iML1515 genome-scale model of *E. coli* metabolism (94). This version of the model contains 1,516 genes, 1,877 metabolites, and 2,712 reactions. All upper and lower bounds were kept the same as in the core model. However, the following changes were made to capture the composition of M9 medium. Lower and upper bound exchange values for ions found in M9 medium ($K^+$, $Mg^{2+}$, $Na^+$, $NH_4^+$, $Cl^-$, $P_i$, $SO_4^{2-}$, and $Ca^{2+}$) were set at −1,000 and 1,000, respectively. The consumption rates of thiamine, glucose, oxygen, and nitrogenous bases were estimated from previously published literature. Parameter justification can be found in the Supplemental Results. Critical lower and upper bound values are in Table S1. For simulations using OptKnock, we sequentially removed all 1,516 genes in the model and considered only genes that increased ATPsyn/biomass above wild type (where no genes are removed) in our analysis. To identify enzymes involved in nucleotide synthesis that increase ATPsyn/biomass, we considered only genes that increased ATPsyn/biomass and that were assigned to one of the following pathways

as reported in Ecocyc: superpathway of histidine, purine, and pyrimidine biosynthesis; superpathway of purine nucleotides *de novo* biosynthesis II; superpathway of guanosine nucleotides *de novo* biosynthesis II; superpathway of pyrimidine deoxyribonucleotides *de novo* biosynthesis, pyrimidine deoxyribonucleotides *de novo* biosynthesis I, and pyrimidine deoxyribonucleotides *de novo* biosynthesis II. To report on flux through purine and pyrimidine synthesis, we simulated the flux activity of the PRAIS and DHORTS reactions, which match the reactions catalyzed by PurM and PyrC, respectively. To simulate the addition of 6-MP, the upper bound value of the GLUPRT, which captures the reaction catalyzed by PurM, was reduced. To simulate the effect of IMP addition, we assigned a lower bound value to the IMP exchange reaction (Ex_IMP_e). See the Supplemental Results for additional information on simulations performed with FBA.

## Growth rate

To determine the growth rate, overnight cultures were washed once in $dH_2O$. One microliter of this culture was diluted into 200 µL of M9 medium contained in an untreated 96-well plate (Genesee Scientific). The medium was overlaid with 70 µL of mineral oil. The plate was placed in a prewarmed (37°C) microplate reader (Victor X4, Perkin Elmer, Waltham, MA), and optical density (OD) at 600 nm ($OD_{600}$) was recorded every 10 minutes for 10 hours. To determine the growth rate, the $OD_{600}$ of cell-free medium was subtracted from all measurements. The maximum growth rate was determined by fitting growth curves to a logistic equation (95) in a custom MATLAB (R2022a, MathWorks Inc., Natick, MA) program as described previously (37). To achieve high-quality fits between the logistic function and growth curves of *ΔpyrC* and *ΔpurK*, we added an extra layer of smoothing by using a median filter with a window size of 2 × 2 to disregard the outliers in the data. In this case, median filtering works better than other filtering methods such as Gaussian filtering and average filtering as a median filter sorts out the neighboring values in the window matrix, chooses the median value, and eliminates the extreme values. Unlike moving average filtering, this method preserves the original data better than other techniques as it uses an existing value in the window instead of numerical averages. Additional details about curve fitting are provided in the Supplemental Methods.

## ATP measurements

ATP measurements occurred as previously described (37, 39). Briefly, an overnight culture was diluted 1/40 into 40 mL of M9 medium with 0.04% glucose but without casamino acids housed in a 50 mL conical tube (Genesee Scientific). The culture was placed in a prewarmed (37°C) shaker incubator and was subsequently shaken at 250 RPM for 2 hours. The bacteria were concentrated twofold in freshly diluted (3:1) M9 medium [containing both 0.04% glucose and casamino acids (0.01%–1%)] using centrifugation. One hundred microliter was subsequently placed in the wells of an opaque walled 96-well microplate (Costar 3370, Corning, Kennebunk, ME), which was overlaid with two Breathe Easy sealing membranes (Sigma Aldrich). The microplate was shaken at 250 RPM at 37°C for 1 hour whereupon $OD_{600}$ reached ~0.1. The BacTiter-Glo assay (Promega, Madison, WI) was used to measure the concentration of ATP according to the manufacturer's recommendations. ATP (luminescence) and $OD_{600}$ were measured in a Victor X4 microplate reader (Perkin Elmer). We used pure ATP (adenosine 5′-triphosphate disodium salt hydrate, Sigma Aldrich) to create a standard curve to quantify the concentration of ATP normalized by cell density ($OD_{600}$) (37). When computing [ATP]/growth rate, we log-transformed ATP so that the order of magnitude was closer to that of growth rate (0.1–1/hour) and to match our previous calculation of growth productivity (37).

## MIC assays

Two hundred microliter of M9 medium containing casamino acids, nitrogenous bases (where indicated), and antibiotics (where indicated) was placed in the wells of an untreated 96-well microplate (Genesee Scientific). An overnight culture was washed

once in $dH_2O$ and diluted either 1,000-fold (high density) or 10,000-fold (low density). Four microliter of these dilutions was inoculated in the growth medium in the 96-well microplate, which was subsequently overlaid with two Breathe Easy sealing membranes (Sigma). All components of the growth medium (e.g., casamino acids, nitrogenous bases, and inhibitors) were added prior to inoculating cells. Cultures were shaken for 24 hours at 250 RPM and at 37°C whereupon the Breathe Easy sealing membranes were removed, and cell density was measured using $OD_{600}$ in a Victor NIVO microplate reader (Perkin Elmer). $OD_{600}$ of cell-free medium was removed from all cell density values. Any condition where growth did not exceed 0.01 was set to zero, as performed previously (Supplemental Results). Concentration gradients used for each antibiotic are found in the Supplemental Methods. The general qualitative trends in $OD_{600}$ are consistent with colony-forming units, and 24 hours is sufficient for MIC assays to reach a steady state, as previously shown (37).

## Reporter assays

Reporter strains were grown overnight with 50 µg/mL of kanamycin. They were then washed twice in $dH_2O$ and diluted 200-fold into M9 medium containing 0.1% casanimo acids (CAA), which was contained in a black-walled 96-well plate (Corning 3340). The medium contained no nitrogenous bases, 5 mM adenine, or 10 mM cytosine. The plate was overlaid with two Breathe Easy sealing membranes (Sigma) and shaken at 37°C and 250 RPM in a darkened incubator. GFP (excite: $OD_{485}$ and emit: $OD_{535}$) and $OD_{600}$ were measured in a Victor X4 microplate reader (Perkin Elmer) after 18 and 24 hours of growth. GFP and $OD_{600}$ values from cell-free medium were subtracted from all measurements, and GFP was normalized by $OD_{600}$. When performing regression analysis between ΔMIC and GFP/$OD_{600}$, we log-transformed GFP/$OD_{600}$ so that it was in the same order of magnitude as ΔMIC (1–25 µg/mL).

## Statistical analysis

The statistical test performed is indicated in the text or figure legend. *t*-Tests (unpaired, unequal variance) were performed in Microsoft Excel (Redmond, WA). Linear and Deming regressions were performed in GraphPad Prism (version 9.3.1, GraphPad, San Diego, CA). For Deming regressions, the average SD in ΔMIC values was used for the error for the y-axis. The average SD for log[ATP]/growth was used for the x-axis. Deming regressions do not report $R^2$ values. WLS regression was performed in JMP Pro 16 (SAS Institute Inc., Cary, NC). We used the inverse variance of ΔMIC values for error on the y-axis. An expansion of errors procedure was used when combining errors from multiple measurements [e.g., the SE of log(ATP)/growth accounts for both the SE from log(ATP) and from growth rate]. ShinyGo (96) (version 0.80) was used to determine fold enrichment and enrichment FDR. Genes whose removal increased ATPsyn/biomass above that of wild type were examined for enrichment against all genes contained within the IML1515 model. We used an FDR cutoff of 0.05, genes were annotated using the Curated.EcoCyc database, and the ecoli_eg_gene ensembl/STRING-db database (captures the *E. coli* K12 – MG1655 genome) was used. For additional information on statistical analysis, see the Supplemental Methods.

## ACKNOWLEDGMENTS

National Institutes of Health grant R15AI159902 (R.P.S. and A.J.L.); National Institutes of Health grant 1R35GM150871-01 (A.J.L.).

Conceptualization: R.P.S. Methodology: D.M.H., M.M., H.I., A.J.L., and R.P.S. Investigation: D.M.H., M.M., M.C., M.C.G., S.R.K., R.C., H.I., T.M., A.R.K., E.M.M., M.B.M., R.T., S.M., S.P., P.M., G.D.T., and R.P.S. Visualization: D.M.H., M.M., M.C., and R.P.S. Supervision: A.J.L. and R.P.S. Writing—original draft: D.M.H. and R.P.S. Writing—review and editing: D.M.H., M.M., M.C., M.C.G., S.R.K., R.C., H.I., T.M., A.R.K., E.M.M., M.B.M., R.T., S.M., S.P., P.M., G.D.T., and R.P.S.

## AUTHOR AFFILIATIONS

[1]Cell Therapy Institute, Kiran Patel College of Allopathic Medicine, Nova Southeastern University, Fort Lauderdale, Florida, USA

[2]Department of Biological Sciences, Halmos College of Arts and Science, Nova Southeastern University, Fort Lauderdale, Florida, USA

[3]Department of Medical Education, Kiran Patel College of Allopathic Medicine, Nova Southeastern University, Fort Lauderdale, Florida, USA

[4]Department of Chemical Engineering, University of Rochester, Rochester, New York, USA

[5]Department of Microbiology and Immunology, University of Rochester Medical Center, Rochester, New York, USA

[6]Department of Biomedical Engineering, University of Rochester Medical Center, Rochester, New York, USA

## AUTHOR ORCIDs

Daniella M. Hernandez http://orcid.org/0009-0003-0299-6921
Allison J. Lopatkin http://orcid.org/0000-0003-0018-9205
Robert P. Smith http://orcid.org/0000-0003-2744-7390

## FUNDING

| Funder | Grant(s) | Author(s) |
| --- | --- | --- |
| HHS \| National Institutes of Health (NIH) | R15AI159902 | Daniella M. Hernandez |
| | | Melissa Marzouk |
| | | Madeline Cole |
| | | Marla C. Fortoul |
| | | Saipranavi Reddy Kethireddy |
| | | Rehan Contractor |
| | | Habibul Islam |
| | | Trent Moulder |
| | | Estefania Marin Meneses |
| | | Maximiliano Barbosa Mendoza |
| | | Ruth Thomas |
| | | Saad Masud |
| | | Sheena Pubien |
| | | Patricia Milanes |
| | | Gabriela Diaz-Tang |
| | | Allison J. Lopatkin |
| | | Robert P. Smith |
| | | Ariane R. Kalifa |
| HHS \| National Institutes of Health (NIH) | 1R35GM150871-01 | Habibul Islam |
| | | Allison J. Lopatkin |

## AUTHOR CONTRIBUTIONS

Daniella M. Hernandez, Investigation, Methodology, Visualization, Writing – original draft, Writing – review and editing | Melissa Marzouk, Investigation, Methodology, Visualization, Writing – review and editing | Madeline Cole, Investigation, Visualization, Writing – review and editing | Marla C. Fortoul, Investigation, Writing – review and editing | Saipranavi Reddy Kethireddy, Investigation, Writing – review and editing | Rehan Contractor, Investigation, Writing – review and editing | Habibul Islam, Investigation, Methodology, Writing – review and editing | Trent Moulder, Investigation, Writing – review and editing | Ariane R. Kalifa, Investigation, Writing – review and editing |

Estefania Marin Meneses, Investigation, Writing – review and editing | Maximiliano Barbosa Mendoza, Investigation, Writing – review and editing | Ruth Thomas, Investigation, Writing – review and editing | Saad Masud, Investigation, Writing – review and editing | Sheena Pubien, Investigation, Writing – review and editing | Patricia Milanes, Investigation, Writing – review and editing | Gabriela Diaz-Tang, Investigation, Writing – review and editing | Allison J. Lopatkin, Methodology, Supervision | Robert P. Smith, Conceptualization, Investigation, Methodology, Supervision, Visualization, Writing – original draft, Writing – review and editing

## DATA AVAILABILITY

All data are available in the main text or the supplemental materials. Raw data and modeling code are also deposited in the Dryad Digital Repository and can be accessed using https://doi.org/10.5061/dryad.931zcrjvt.

## ADDITIONAL FILES

The following material is available online.

### Supplemental Material

**Supplemental material (Spectrum01895-24-S0001.pdf).** Fig. S1 to S20; Tables S1 to S10; Methods.

### Open Peer Review

**PEER REVIEW HISTORY (review-history.pdf).** An accounting of the reviewer comments and feedback.

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
