## [Reviewer comments · Microbiology Spectrum]

Microbiology Spectrum

Purine and pyrimidine synthesis differently affect the strength of the inoculum effect for aminoglycoside and β -lactam antibiotics.

Daniella Hernandez, Melissa Marzouk, Madeline Cole, Marla Fortoul, Saipranavi Reddy Kethireddy, Rehan Contractor, Habibul Islam, Trent Moulder, Ariane Kalifa, Estefania Marin Meneses, Maximiliano Barbosa Mendoza, Ruth Thomas, Saad Masud, Sheena Pubien, Patricia Milanés, Gabriela Diaz-Tang, Allison Lopatkin, and Robert Smith

Corresponding Author(s): Robert Smith, Nova Southeastern University

Review Timeline:

Submission Date:	August 12, 2024
Editorial Decision:	September 16, 2024
Revision Received:	September 20, 2024
Accepted:	September 24, 2024

Editor: Brian Conlon

Reviewer(s): Disclosure of reviewer identity is with reference to reviewer comments included in decision letter(s). The following individuals involved in review of your submission have agreed to reveal their identity: Peter Belenky (Reviewer #1); Jinki Yeom (Reviewer #2)

Transaction Report:

DOI: <https://doi.org/10.1128/spectrum.01895-24>

Re: Spectrum01895-24 (Purine and pyrimidine synthesis differently affect the strength of the inoculum effect for aminoglycoside and β -lactam antibiotics.)

Dear Dr. Robert P Smith:

Thank you for the privilege of reviewing your work. Below you will find my comments, instructions from the Spectrum editorial office, and the reviewer comments.

Please see reviewer 2 comments on statistical analyses.

Revision Guidelines

Sincerely,
Brian Conlon
Editor
Microbiology Spectrum

Reviewer #1 (Comments for the Author):

The study "Purine and pyrimidine synthesis differently affect the strength of the inoculum effect for aminoglycoside and β -lactam antibiotics" is a significant contribution to understanding the inoculum effect (IE) in bacterial populations. By integrating computational modeling with experimental data, the authors demonstrate that purine and pyrimidine biosynthesis pathways play a key role in determining the strength of the IE for aminoglycosides and β -lactams. This work is particularly strong in its

methodological approach, using both Flux Balance Analysis and Optknock, and in its relevance for antibiotic resistance research. The findings highlight the potential to manipulate nucleotide synthesis pathways to reduce the inoculum effect, paving the way for novel therapeutic strategies in treating high-density bacterial infections.

The authors provided thorough, well-supported responses, expanding on experimental details, addressing conceptual concerns, and making revisions to clarify key points in the manuscript. In most cases, they either adjusted the manuscript to address the reviewers' comments or provided solid reasoning for why certain choices were made (e.g., explaining the rationale behind their antibiotic choices). Their responses appear adequate, particularly given that the majority of the reviewers' comments were either requests for clarification or minor points of criticism. The manuscript adjustments made in response to the reviews seem to strengthen the overall clarity and rigor of the work.

Reviewer #2 (Comments for the Author):

The researchers propose that the synthesis of purines and pyrimidines has an antibiotic class-specific impact on the intensity of the inoculum effect, which in turn influences antibiotic effectiveness. Earlier research demonstrated that the correlation between ATP concentration and growth rate can explain the strength and occurrence of the inoculum effect for bactericidal antibiotics.

Through a combination of flux balance analysis and experimental work, the study shows that nucleotide synthesis can determine the relationship between ATP concentration and growth rate, thereby affecting the strength of the inoculum effect in a manner that depends on the class of antibiotic. When the ratio of ATP concentration to growth rate is sufficiently high, as influenced by externally provided nitrogenous bases, the inoculum effect does not occur. This finding holds true for both *Escherichia coli* and *Pseudomonas aeruginosa*.

The study underscores the antibiotic class-specific influence of purine and pyrimidine synthesis on the severity of the inoculum effect. This insight may lead to the development of strategies to mitigate the inoculum effect in clinical settings.

This manuscript is very clear to me, but minor aspects of the manuscript necessitate revisions.

1. In many figures, they are missing statistic analysis. For example, In figure2, panel C has statistic analysis with asterisk mark, but panel D does not have it.

We want to thank both reviewers for their time and expertise in reviewing our original submission. Please find a point-by-point response to their comments below in blue.

Reviewer #1 (Comments for the Author):

The study "Purine and pyrimidine synthesis differently affect the strength of the inoculum effect for aminoglycoside and β -lactam antibiotics" is a significant contribution to understanding the inoculum effect (IE) in bacterial populations. By integrating computational modeling with experimental data, the authors demonstrate that purine and pyrimidine biosynthesis pathways play a key role in determining the strength of the IE for aminoglycosides and β -lactams. This work is particularly strong in its methodological approach, using both Flux Balance Analysis and Optknock, and in its relevance for antibiotic resistance research. The findings highlight the potential to manipulate nucleotide synthesis pathways to reduce the inoculum effect, paving the way for novel therapeutic strategies in treating high-density bacterial infections.

The authors provided thorough, well-supported responses, expanding on experimental details, addressing conceptual concerns, and making revisions to clarify key points in the manuscript. In most cases, they either adjusted the manuscript to address the reviewers' comments or provided solid reasoning for why certain choices were made (e.g., explaining the rationale behind their antibiotic choices). Their responses appear adequate, particularly given that the majority of the reviewers' comments were either requests for clarification or minor points of criticism. The manuscript adjustments made in response to the reviews seem to strengthen the overall clarity and rigor of the work.

We would like to thank the reviewer for the kind words and positive assessment of the manuscript.

Reviewer #2 (Comments for the Author):

The researchers propose that the synthesis of purines and pyrimidines has an antibiotic class-specific impact on the intensity of the inoculum effect, which in turn influences antibiotic effectiveness. Earlier research demonstrated that the correlation between ATP concentration and growth rate can explain the strength and occurrence of the inoculum effect for bactericidal antibiotics.

Through a combination of flux balance analysis and experimental work, the study shows that nucleotide synthesis can determine the relationship between ATP concentration and growth rate, thereby affecting the strength of the inoculum effect in a manner that depends on the class of antibiotic. When the ratio of ATP concentration to growth rate is sufficiently high, as influenced by externally provided nitrogenous bases, the inoculum effect does not occur. This finding holds true for both *Escherichia coli* and *Pseudomonas aeruginosa*.

The study underscores the antibiotic class-specific influence of purine and pyrimidine synthesis on the severity of the inoculum effect. This insight may lead to the development of strategies to mitigate the inoculum effect in clinical settings.

We would like to thank the reviewer for their positive assessment of our manuscript.

This manuscript is very clear to me, but minor aspects of the manuscript necessitate revisions.

1. In many figures, they are missing statistic analysis. For example, In figure2, panel C has statistic analysis with asterisk mark, but panel D does not have it.

We thank the reviewer for their thorough assessment of our data. The reason that panel D does not have a statistical analysis is that it presents changes in ATP/growth rate. This metric is the average of independent biological replicates of ATP and growth rate measured on different days from different starting colonies. To achieve this metric, we averaged ATP and growth from these experiments and achieved a single value. The SEM presented in calculated using an expansion of errors approach.

However, we cannot find a widely used/accepted statistical approach that can perform statistical analysis when the average of multiple independent experiments is used to generate a single value; in general, and to our best knowledge, single values cannot be used in commonly performed statistical analyses. We could, of course, take individual [ATP] values and divide them by individual growth rate values. But because each [ATP] and growth rate value is determined from completely independent experiments, which [ATP] and growth rate value we use to generate each [ATP]/growth rate would be completely arbitrary. Finally, because we have P values generated for each [ATP] and growth rate value when they are compared amongst one another, we cannot find an analysis that would take this into account when generating [ATP]/growth rate. Because of these caveats, we chose to err on the side of caution and not make any claims of significance on [ATP]/growth rate relative to one another. The omission of P values for [ATP]/growth rate is consistently applied throughout the entire manuscript.

Instead, we chose to highlight the trends and only compare changes in Δ MIC relative to ATP/growth rate. We have performed this using linear, Deming, and weight least squares regression. Importantly Deming regressions account for error on the x-axis (in this case ATP/growth rate); thus we are indeed using the error on these values to test the significance of the relationship between Δ MIC and [ATP]/growth rate, which is the central focus of the manuscript.

We did not supply statistical analysis for the simulations as they are deterministic and report only single values based on a set of parameters. To perform statistical analysis, all simulations would need to be converted to stochastic simulations; we do not know of a widely available flux balance analysis program that can provide these types of simulations at the population level.

We have noted the above by indicating the following in a new section in the Supplemental Methods. We hope that this avoids confusion in the future:

“We did not perform statistical analysis on ATPsyn, biomass, and additional flux values because FBA simulations are deterministic and only report a single set of values for a given parameter set. We did not perform a statistical analysis on [ATP]/growth rate values as both [ATP] and growth rate were measured from different biological replicates on different days. Therefore, they cannot be immediately paired to formulate a single [ATP]/growth rate value for each biological replicate that can be used for statistical analysis. Moreover, we could not find a well-established statistical analysis method that can account for the above while accounting for significant differences amongst [ATP] and growth rate values. Accordingly, we did not perform a statistical analysis on [ATP]/growth rate

values throughout the manuscript. We do, however, test the significance of the error on [ATP]/growth relative to \square MIC using Deming regressions, as noted throughout the manuscript.” In terms of the rest of the manuscript. We did not report P values raw OD₆₀₀ data (e.g., Figs. S5, S6), [ATP] at each percentage of casamino acids (e.g., Fig. S4), or growth rate each percentage of casamino acids (e.g., Fig. S2) because we do not make any claims of significance in any one of these sets of data, nor are they critical to the core claims and conclusion in the manuscript. However, because we are supplying all of our raw data on Dryad, if the community wishes to assess the significance of these values, they are more than welcome to.

Our approach to growth curve fitting does not report a statistical value; instead, we reported a goodness of fit using the residual values (e.g., Table S2).

Otherwise, in any instance where a statistical claim is made, the P value is presented in the main text (e.g., Figure 1) or in the figure legend (e.g., Figure 2). Where multiple values are significant from each other, we indicated in the figure legend $P \leq 0.xyz$, where xyz represents the highest P value that continued to demonstrate significance in the figure panel. We also reported all P values in this instance in multiple tables in the supplement. For example, all P values that pertain to claims of significance in Figure 2 can be found in Fig. S2. We chose this approach so that our figure legends were brief and so that any reader could access our statistical analysis in the Supplement.

Overall, we contend that in instances where significance is claimed, the P values have been reported in the figure legend or the tables found in the supplemental material.

Re: Spectrum01895-24R1 (Purine and pyrimidine synthesis differently affect the strength of the inoculum effect for aminoglycoside and β -lactam antibiotics.)

Dear Dr. Robert P Smith:

Your manuscript has been accepted, and I am forwarding it to the ASM production staff for publication. Your paper will first be checked to make sure all elements meet the technical requirements. ASM staff will contact you if anything needs to be revised before copyediting and production can begin. Otherwise, you will be notified when your proofs are ready to be viewed.

Sincerely,
Brian Conlon
Editor
Microbiology Spectrum